# Surface Functionalization of Nanocarriers with Anti-EGFR Ligands for Cancer Active Targeting

**DOI:** 10.3390/nano15030158

**Published:** 2025-01-21

**Authors:** Alessandra Spada, Sandrine Gerber-Lemaire

**Affiliations:** Group for Functionalized Biomaterials, Institute of Chemical Sciences and Engineering, Ecole Polytechnique Fédérale de Lausanne, 1015 Lausanne, Switzerland; alessandra.spada@epfl.ch

**Keywords:** active targeting, EGFR, biomolecules, targeting ligands, targeted therapy, nanomedicine, cancer targeting

## Abstract

Active cancer targeting consists of the selective recognition of overexpressed biomarkers on cancer cell surfaces or within the tumor microenvironment, enabled by ligands conjugated to drug carriers. Nanoparticle (NP)-based systems are highly relevant for such an approach due to their large surface area which is amenable to a variety of chemical modifications. Over the past decades, several studies have debated the efficiency of passive targeting, highlighting active targeting as a more specific and selective approach. The choice of conjugation chemistry for attaching ligands to nanocarriers is critical to ensure a stable and robust system. Among the panel of cancer biomarkers, the epidermal growth factor receptor (EGFR) stands as one of the most frequently overexpressed receptors in different cancer types. The design and development of nanocarriers with surface-bound anti-EGFR ligands are vital for targeted therapy, relying on their facilitated capture by EGFR-overexpressing tumor cells and enabling receptor-mediated endocytosis to improve drug accumulation within the tumor microenvironment. In this review, we examine several examples of the most recent and significant anti-EGFR nanocarriers and explore the various conjugation strategies for NP functionalization with anti-EGFR biomolecules and small molecular ligands. In addition, we also describe some of the most common characterization techniques to confirm and analyze the conjugation patterns.

## 1. Introduction

Cancer mortality is reduced when cases are detected and treated at an early stage of development. However, even though much progress has been made in the management of patients, there are still issues that need to be addressed to improve cancer early diagnosis and provide innovative therapeutic modalities [1]. Conventional imaging techniques and cancer treatments lack sensitivity for early detection and show poor specificity, causing adverse side effects. In order to overcome some of their limitations, nanomedicine has emerged as a promising alternative in the past decades [2].

Nanoparticles (NPs) have attracted interest due to their unique features. They are very small colloidal systems, with sizes in the nanometer range, whose morphology and properties depend on their compositions. NPs utilized for therapeutic and diagnostic applications can be classified into several categories, including polymeric, inorganic and lipid-based NPs. Polymeric NPs encompass systems such as polymersomes [3], dendrimers [4] and nanomicelles [5]. Inorganic NPs include mesoporous silica NPs [6], iron oxide NPs [7], gold NPs [8] and quantum dots [9]. Lipid-based NPs primarily consist of liposomes [10] and lipid NPs (LNPs) [11]. Polymeric NPs typically exhibit mean diameters ranging from 100 to 300 nm. They offer advantages such as precise control over particle characteristics, easy surface functionalization, controlled release capabilities, and protection of drug and payloads from environmental degradation. However, they are prone to aggregation and may induce toxicity in certain cases [12]. Inorganic NPs present high variability in size, structure and geometry [13], with specific electrical, magnetic and optical properties, making them highly suitable for theranostic applications [14]. Nonetheless, they can exhibit toxicity if not coated with biocompatible layers [15]. Lipid-based NPs provide benefits such as enhanced solubility and bioavailability of hydrophobic payloads, controlled release, and biocompatibility [11,16]. Conversely, their drug loading efficiency is lower compared to other types of NPs [17]. NPs are characterized by a high surface-to-volume ratio that generally allows high loading of therapeutic agents and/or imaging probes [18]. To be used in biomedical applications, nanomaterials need to be biocompatible, well characterized and stable in vivo [19]. As compared to traditional chemotherapeutic agents, NPs can encapsulate hydrophobic molecules, increasing their solubility, biocompatibility and retention time at tumor sites [20]. They are capable of co-delivering drug resistance inhibitors while enabling controlled and sustained drug release [21], and circumvent the drug efflux mechanisms, leading to decreased multidrug resistance (MDR) [22]. The surface characteristics of NPs determine their bioavailability and half-life [23]. For example, one of the main obstacles of NP delivery is the opsonization (coating of the NPs by non-specific proteins that leads to immune recognition), sequestration by the reticuloendothelial system (RES) and clearance by the immune system [24]. Smaller NPs (<200 nm) have been reported to escape the RES more efficiently [25,26,27,28]. In addition, NPs coated with hydrophilic materials, such as polyethylene glycol (PEG), have been shown to reduce opsonization, thus increasing their circulation time and improving their penetration and accumulation in tumors [28,29]. To improve the targeting of loaded therapeutic agents, enhance efficiency and minimize adverse side effects, dynamically responsive nano-drug delivery systems (DRNSs) have been explored as advanced tools capable of directional drug release with self-regulation and self-feedback mechanisms in response to specific stimuli [30].

The ability of NPs to target malignant cells and tumor tissues results from either passive or active targeting, or a combination of both [31]. The main driving force for passive targeting is the so-called enhanced permeation and retention effect (EPR) [32,33], caused by the leaky tumor vasculature, first reported by Matsumura and Maeda in 1986 [34]. The tumor vascularization presents large pores in the vascular wall and the NPs tend to leak from the blood vessels and accumulate within tumor tissues. At the same time, poor tumor lymphatic drainage increases the retention of the NPs [33]. In the EPR effect, these unique anatomical-pathophysiological features of tumor vasculature allow the transport and internalization of NPs into tumor tissues [35]. Several studies demonstrated the relation between the EPR effect and the size of the NPs: smaller NPs have shown better penetrability into tumor tissues [36,37].

However, there are several limitations with regard to passive targeting, including non-specific NP distribution, non-universal existence of the EPR effect and different permeability of blood vessels across various tumors. In the past decades, the efficiency of passive targeting has been debated, since the model was proved to be oversimplified [38], accounting for only 0.7% of the uptake by tumor tissues, due to non-specific uptake by healthy organs [39,40].

Active targeting is intended to increase the concentration of nanocarriers and their bioavailability at tumor sites [41]. This strategy relies on the specific recognition of biomarkers which are homogeneously overexpressed by cancer cells or cancer associated cells [42]. It mostly addresses tumor cells, but can also be directed toward neighboring angiogenic endothelial cells, or toward the mildly acidic tumor microenvironment. Therefore, as opposed to passive targeting, active targeting requires the grafting of biomarker-specific ligands to the surface of the nanocarriers, thus increasing the selective recognition of tumor cells (Figure 1) [43].

Several receptors are overexpressed or specifically expressed in different types of cancer, such as folate receptors, integrins, transferrin receptors, G protein-coupled receptors, sigma receptors, fibroblast growth factors and epidermal growth factor receptor (EGFR) [44]. In particular, EGFR is overexpressed in several types of cancer, including non-small cell lung, bladder, gastric, kidney, ovarian, colorectal, breast, pancreatic cancers and squamous-cell carcinoma of head and neck cancers [45,46,47]. For this reason, nano delivery systems decorated with EGFR-targeting ligands have gained increasing attention as potential tools to achieve enhanced specificity through receptor-mediated endocytosis. In addition, there has been a number of significant studies highlighting the benefits of the combination between EGFR targeting and other treatment modalities, including immunotherapy [48,49]. For example, a bispecific antibody (BsAb) that simultaneously targets both EGFR and programmed cell death protein 1 (PD1), a protein that is involved in immune checkpoint blockade, has been investigated as a novel and promising strategy to effectively treat cancers [50,51]. In addition to the EGFR targeting, the BsAb exhibits a potent antibody-dependent cellular cytotoxicity (ADCC) activity and activates antitumor immunity through blockade of PD1/PD-L1 interaction. Moreover, it was demonstrated that the BsAb is more potent than the individual monoclonal antibodies (mAbs) and their combination at targeting and inhibiting tumor growth. Liposomal nanohybrid cerasomes decorated with anti-EGFR mAb and PD-L1 mAb were explored for targeted tumor imaging and photodynamic therapy (PDT) [52]. The experiments showed that PDT with the EGFR-targeted system combined with PD-L1 mAb treatment strategy was more effective against tumors in comparison with the simultaneous non-targeted delivery of PDT with liposomal nanohybrid cerasomes and PD-L1 mAb treatment. In another study, simultaneous targeting of EGFR and CD73, an emerging checkpoint for cancer immunotherapy, was investigated as a new therapeutic approach for breast cancer [53]. The combined treatment showed a significant effect in inhibiting the growth and spread of the tumors.

The choice of the conjugation strategy for the grafting of ligands to nanocarriers is a key parameter to ensure their stability and reliability, and is dictated by diverse factors, such as the physicochemical parameters of the nanocarriers (including size, composition, surface charge), the type and composition of the selected ligands. Covalent conjugation refers to the use of direct or spaced covalent bonds, while non-covalent immobilization relies on surface adsorption through electrostatic interactions, hydrophobic interactions, hydrogen bonding and similar types of weaker forces (Figure 2) [54]. Covalent bonds are more stable and resistant, but they often require the introduction of chemical modifications on both the surface of the nanocarriers and ligands. Thus, NPs need to be precisely engineered for the presentation of surface functional groups (thiols, carboxylic acids, amines) that can be exploited as chemical handles for further post-functionalization. While being easy to produce, non-covalent systems are often reversible and suffer from low stability in physiological environments [55].

Previous reviews described the large number of biological ligands which were identified for facilitating active targeting [31,41,43,56,57,58]. However, only few reports focused on the conjugation chemistry and the means of characterization of the functionalized NPs.

We herein highlight the most recent and significant anti-EGFR functionalized nanocarriers with a focus on the last five years of the literature. Part of the review also presents the characterization techniques that are commonly used to assess the presence of ligands on the NP surface.

## 2. Epidermal Growth Factor Receptor

EGFR, also known as ErbB1 or HER1, is a 170 kDa glycoprotein composed of 1186 amino acid residues, and was discovered by Nobel Prize-winning Cohen and colleagues in 1978 [59]. It belongs to the human epidermal growth factor receptor (HER) family of four closely related receptor tyrosine kinases (RTKs). The other components of the family, ubiquitously expressed in epithelial, cardiac, neuronal and mesenchymal cells, are HER2 (ErbB2), HER (ErbB3), and HER4 (ErbB4). The HER family plays vital roles in the modulation of processes in healthy cells, including cell proliferation, motility, survival and differentiation [60], ensuring that the kinetics of these phenomena correspond to the tissues’ requirement for homeostasis. It was shown that any dysregulation of these processes leads to cancer development, making therefore EGFR one of the main anticancer targets [61].

As all other members of the HER family, EGFR is a transmembrane protein, consisting of an extracellular domain (ECD), where the ligand binding site resides, a transmembrane domain (TMD), and an intracellular tyrosine kinase domain, which houses the catalytic activity of the receptor [62]. The ECD of EGFR is composed of two homologous domains, DI and DIII, involved in the ligand binding, and DII and DIV, that form disulfide bonds [63]. In an inactive state, the ECD has a tethered configuration, presenting intra-molecular bondages blocking the dimerization arm. Upon ligand binding to the ECD, the cytoplasmatic domain undergoes dimerization and phosphorylation, with the kinase activation being allowed by the configurational changes [64]. The EGFR activation leads to a cascade of subsequent downstream signal transduction pathways, resulting in the mitogenic and anti-apoptotic signal cascades [65]. Among those, the most important are the phosphatidylinositol-3-kinase (PI3K) signaling network, responsible for tumor growth, the RAS/MAPK pathway and the JAK2/STAT pathway (Figure 3) [66].

It was established that the EGFR pathway can be activated by several mechanisms, including receptor mutation in specific domains, overexpression, inefficient inactivation or augmented ligand production [67]. Some of these processes, such as genetic alterations and increased ligand production, occur simultaneously due to autocrine or paracrine loops [68,69]. Furthermore, EGFR genetic mutations lead to abnormal EGFR trafficking, contributing to increased signaling and tumor growth. In normal cells, the number of EGFR is estimated to be around 40,000–100,000 receptors per cell [70], whereas in cancer cells this number rises to more than 10^6^ receptors per cell [71]. Several studies established the interconnection between EGFR phosphorylation and carcinogenic events, such as smoke inhalation, exposure to ultraviolet radiation and bacterial infections. Consequently, malignant tumors are more likely to develop in tissues where the receptor is found [65]. EGFR overexpression was observed in a variety of human solid tumors, such as kidney, pancreas, breast, ovary, bladder, colorectal, head and neck and lung cancers [72]. The EGFR levels can be assessed through receptor quantification at the DNA, RNA or protein level, or through investigation of the degree of signaling by studying the activation of the receptor or the downstream markers [72]. For these purposes, different techniques are used, such as DNA analysis, northern blotting or quantitative reverse transcription polymerase chain reaction (rtPCR), immunohistochemistry (IHC), western blot analysis and enzyme immunoassay.

Due to its central roles in a wide range of key cellular processes, EGFR was selected in many targeting and inhibition strategies [73]. The most clinically advanced EGFR inhibitors include mAbs blocking the EGFR’s ECD, and small-molecular inhibitors of the intracellular tyrosine kinase domain [74]. To date, fourteen EGFR-targeting agents have been approved for cancer treatments. The Food and Drug Administration agency (FDA) approved several anti-EGFR mAbs, among which cetuximab (2004), panitumumab (2006), necitumumab (2015), amivantamab (2021) (The Antibody Society. Therapeutic monoclonal antibodies approved or in review in the EU or US; www.antibodysociety.org/resources/approved-antibodies; accessed on 25 November 2024). Many other candidates are undergoing clinical studies, such as nimotuzumab, ZZ06, JMT101, SCT200, HLX07 (clinicaltrials.gov), mAb806 [75]. By binding to the ECD and preventing the dimerization processes, these agents are designed to selectively target and kill tumor cells. Tyrosine kinase inhibitors (TKIs), such as erlotinib, gefitinib, lapatinib and icotinib (first-generation), afatinib, neratinibs and dacomitinib (second-generation), or osimertinib (third-generation), target the intracellular tyrosine kinase domain, competing with ATP for binding, and blocking the proliferation signaling [76]. Among the TKIs, gefitinib, erlotinib, lapatinib, afatinib, brigatinib, osimertinib, dacomitinib, vandetanib have received FDA approval for clinical use in cancer therapy [77]. However, the clinical application of EGFR-targeting ligands remains hindered by substantial challenges. A major limitation is the development of resistance to treatment, which impedes the optimal efficacy of EGFR inhibitors. While some patients develop de novo resistance to EGFR inhibition and fail to respond to therapy, others who initially responded to therapy eventually develop acquired resistance [78]. Tumor cells can develop resistance to EGFR inhibitors through various mechanisms, including the acquisition of secondary mutations in the EGFR gene. Additionally, they may activate alternative signaling pathways that circumvent EGFR dependency to sustain cell proliferation or upregulate other receptor tyrosine kinases to compensate for EGFR inhibition [79]. Intratumoral heterogeneity is another well-recognized contributor to resistance against targeted therapies and is considered a major factor in treatment failure [80]. Heterogeneity of the EGFR protein expression has been observed in clinical samples from multiple cancer types [81,82,83,84]. Furthermore, the clinical use of EGFR inhibitors is often limited by adverse side effects, such as skin rashes, diarrhea and hepatotoxicity [85]. Future directions and emerging strategies to bridge the gap between EGFR-targeted therapy research and its clinical translation include the development of next-generation inhibitors, designed to overcome resistance mutations, the implementation of combination therapies, that address both genetic and non-genetic mechanisms of resistance, and the identification of predictive biomarkers, essential for guiding patient selection, optimizing treatment strategies and improving the overall efficacy of EGFR-targeted therapy [86,87].

In addition to its relevance as a therapeutic target, EGFR stands among the most targeted receptors for cancer active targeting [58]. This review focuses on anti-EGFR targeting ligands-functionalized NPs, which are therefore specific to EGFR-overexpressing tumors.

## 3. EGFR-Targeting Ligands

This section focuses on the description of different nanocarriers functionalized with the main anti-EGFR ligands, such as peptides, proteins, polysaccharides, nucleic acids and small molecules. Each family of anti-EGFR targeting agents is described, highlighting their performance and limitations.

### 3.1. Epidermal Growth Factor

Epidermal growth factor (EGF) is a small protein, constituted by only 53 amino acids with a molecular weight of approximately 6 kDa. Present in different mammalian species, it was first isolated from parotid gland of male mice and subsequently, human EGF (hEGF) was purified from human urine [88]. hEGF is known as “urogastrone” for its ability to inhibit gastric acid secretion in humans [89,90]. The protein structure presents six cysteine residues able to form three internal disulfide bonds, that can be used for further NP conjugation [91]. The EGF protein possesses a very high affinity for EGFR, with a dissociation constant (Kd) of 2 nM. In 1986, Cohen et al. established that the protein directly stimulated the proliferation of epidermal cells, and that the stimulatory action did not depend on other systemic or hormonal influences [92]. In addition, hEGF has various effects on cell regeneration, including migration of keratinocytes, formation of granulation tissues and stimulation of fibroblast motility, which play major roles in wound healing processes [93,94]. EGF exerts its effects in the target cells by binding to the plasma membrane located EGFR [95].

The advantages of using the EGF protein as targeting agent include: (i) smaller sizes compared to other targeting ligands such as full antibodies (Abs, 6 kDa vs. 150 kDa, respectively); (ii) lower cytotoxicity being one of the native ligands of EGFR [96]; (iii) ease of conjugation to nanocarriers thanks to the presence of disulfide bonds and hydrophobic regions in its structure [97]; (iv) stability at physiological conditions and neutral pH due to an isoelectric point (pI) of around 4.5, making it negatively charged at neutral pHs [98]. On the other hand, EGF production is expensive, less convenient to obtain from human resources, and can cause antigenicity issues when obtained from murine sources.

Zhang et al. demonstrated the importance of the ligand orientation when conjugating EGF to the surface of NPs, in order to minimize interference in the EGF-EGFR interaction [99]. EGF was conjugated to NPs’ surface through covalent functionalization via amide bond using the primary amino groups of EGF. However, the presence of three amino groups (N-terminus, Lys48 and Lys28) in the EGF structure did not allow for site-directed chemical coupling. Furthermore, Lys48 and Lys28 are located close to the region where the protein interacts with its receptor. The conjugation was therefore limited to the N-terminus of EGF, by producing three lysine-free EGF, modifying lysine (K) with arginine (R) or serine (S) using gene recombination technology. Among the tested formulations (RS-, SR-EGF-GNPs conjugates), the second one showed enhanced biological activities and growth inhibition in EGFR-overexpressing skin cancer cell line A431. This effect was demonstrated to be due not only to the orientation control, but also to increased binding activities of this mutant EGF to EGFR.

Castilho et al. developed a treatment modality for triple-negative breast cancer, using bifunctional theranostic nanoprobes (BN) for PDT on human breast carcinoma and normal human cells [100]. BNs were modified with EGF targeting ligand and chlorin e6 (Ce6). Conjugation to AuNPs led to 10-fold increased efficacy of the nanoplatform compared with free Ce6. The conjugates induced triple negative breast cancer (TNBC) cell death by increasing reactive oxygen species (ROS) levels, while they did not impact normal cells from human breast epithelium.

Salama et al. engineered EGF-ligated polyethylene glycol-coated TiO_2_ NPs (EGF-TiO_2_ PEG NPs) to overcome the low cellular uptake and cell-proliferating ability of TiO_2_ NPs for PDT and photodynamic diagnosis (PDD) [101]. On A431 epidermal cancer cell line, the binding of EGF-TiO_2_ PEG NPs to EGFR induced receptor-mediated endocytosis, leading to increased NP cellular uptake, decreased localization of EGFR on the cell surface, and decreased signaling for cell proliferation compared with unconjugated NPs (Figure 4).

Another example of EGF-functionalized AuNPs for photothermal therapy (PTT) consisted of multifunctional hybrid nanocarriers combined with near-infrared (NIR) laser irradiation for the treatment of melanoma [102]. AuNPs were coated with hyaluronic (HA) and oleic acid (OA) and functionalized with EGF to improve the in vivo efficiency in hairless immunocompromised mice. These cells overexpress multiple receptors, including CD44 and EGFR, which were targeted by HA and EGF, respectively. In vivo experiments with EGF-conjugated HAOA-coated GNPs and NIR laser using different exposure times showed a significant reduction in the volume of melanoma tumor 24 h post-treatment (up to 81% reduction), whereas the control group did not show any significant tumor volume change. Moreover, the nanocarriers did not affect the normal function of organs and did not induce any inflammatory systemic response in the 24 h period post-administration.

Susnik et al. studied the effect of EGF conjugation on cell signaling, modulation of endocytic activity, and the uptake of different SiO_2_ NPs (59 and 422 nm) and PEGylated AuNPs in A549 lung epithelial carcinoma cells [103]. Cell stimulation with EGF facilitated the uptake of 59 nm fluorescently labelled SiO_2_-BDP FL NPs, in particular under simultaneous and sequential co-exposure scenario. In contrast, reduced uptake of larger NPs was observed, probably due to the differences in F-actin dynamics during endocytosis depending on the NP sizes. In addition, the cellular uptake of EGF-conjugated PEGylated AuNPs carrying antisense oligonucleotides (ASO), complementary to the *c-MYC* transcript (Au@PEG@c-myc), was enhanced in A549 cells and the *c-MYC* silencing efficiency was demonstrated.

Other relevant studies disclosed the conjugation of EGF to AuNPs [104,105,106], liposomes [107], polymeric NPs [108,109,110,111], and lipid-polymer hybrid NPs [112] for anticancer therapy.

### 3.2. GE11 Peptide

Peptides have increasingly been investigated as targeting agents for cancer active targeting. They are characterized by low immunogenic potential and high penetration capacity into solid tumors. Furthermore, peptides can be easily synthesized, in particular through microwave-assisted solid-phase automated synthesis [113], and conjugated to nanocarriers through multiple functionalization techniques [114]. Multiple EGFR-binding peptides were selected from phage display peptide libraries, such as the dodecapeptide GE11 (YHWYGYTPQNVI). Even if characterized by a lower affinity for EGFR compared with EGF by an order of magnitude (Kd of 22 nM vs. Kd of 2 nM, respectively), higher surface densities of peptide can be achieved, leading to enhanced targeting efficiency [115]. Furthermore, GE11 is exempt from mitogen activity and presents a significantly reduced size compared with EGF or mAbs. Previous studies showed a high potential of GE11 to facilitate NP endocytosis, probably due to an alternative EGFR-dependent actin-driven pathway [115].

Guo et al. developed an EGFR-targeted multifunctional micellar nanoplatform by encapsulating celecoxib and doxorubicin (DOX) into polymeric micelles based on the conjugate of GE11-PEG-b-poly(trimethylene carbonate) to suppress metastatic breast cancer proliferation and metastasis [116]. GE11 conjugation resulted in enhanced tumor-targeted accumulation and cellular uptake. Compared with negative controls (micelles without GE11 or delivering only DOX), systemic administration of the targeted platform into mice bearing subcutaneous 4T1 tumor models led to higher tumor growth suppression and decreased lung metastasis.

Alternatively, ^64^Cu-labeled GE11-decorated polymeric micellar NPs (PMNPs) were formed through self-assembly of poly(ethylene oxide) (PEO)-*block*-poly(ester)s, based on PEO-*b*-poly(α-benzyl carboxylate-ε-caprolactone) (PEO-b-PBCL) [117]. In comparison with non-targeted PMNPs, GE11-conjugated nanocarriers showed a significantly higher internalization into EGFR-overexpressing colorectal cancer cells HCT116 (Figure 5) and higher tumor uptake 24 h post-injection. In vivo PET data revealed a 28% increase in tumor accumulation and retention of the radiolabeled GE11-decorated NPs.

A later report focused on GE11-conjugated PMNPs carrying A83B4C63, a novel inhibitor of polynucleotide kinase/phosphatase (PNKP) in colorectal cancer (CRC) [118]. GE11 improved the activity of A83B4C63 in EGFR+ CRC cells in vitro. In vivo, the nanoplatform showed a trend toward increased primary tumor homing in an orthotopic CRC xenograft, even though statistical significance for the accumulation of GE11-modified micelles compared with plain ones was not reached.

Du et al. synthesized salinomycin-loaded GE11-conjugated polymer-lipid hybrid NPs (GE11-NPs-SAL) to target osteosarcoma [119]. The constructs inhibited the migration and proliferation of EGFR-overexpressing U2OS osteosarcoma cells and induced enhanced tumor growth reduction in vivo, compared with non-targeted micelles. GE11-targeted delivery of SAL to breast cancer cells was also achieved with poly-lactic-co-glycolic acid (PLGA)/tocopheryl polyethylene glycol succinate (TPGS) NPs [120]. In vitro flow cytometry assays showed that GE11-SAL NPs were internalized to a higher extent in MCF-7 cells compared with non-targeted SAL NPs. In vivo, GE11-SAL NPs displayed a marked reduction in tumor volume up to 30 days in BALB/C nude mice bearing MCF-7 breast cancer xenografts.

Other studies explored the functionalization of organic nanocarriers, particularly liposomes, with GE11. Tang et al. developed GE11-decorated PEGylated liposomes (GE11-TLs) that exhibited higher retention and uptake in EGFR-overexpressing stromal cells in SMMC-7721 xenograft models [121]. Furthermore, the functionalized nanoplatform reduced the intravasation of liposomes back into blood vessels, enhancing tumor-specific delivery. Zhou et al. developed GE11-installed chimeric polymersomes (GE11-CPs) for EGFR-targeted protein therapy in SMMC-7721 tumor models [122]. Saporin-loaded GE11-CPs (with 10% GE11) showed over 3-fold enhanced uptake in SMMC-7721 cells and significantly increased anticancer potency compared with non-targeted controls. In vivo, biodistribution studies revealed 3-fold higher tumor accumulation of the functionalized nanoplatform.

Additional studies focused on the combination of GE11 targeting and NIR irradiation for targeted therapeutic applications. For example, Huang et al. developed GE11-decorated liposomes encapsulating curcumin and indocyanine green for synergistic cancer therapy (GE11-CUR/ICG-LPs) [123]. In EGFR-overexpressing A549 cells, NIR irradiation triggered hyperthermia for tumor ablation while releasing curcumin to eliminate residual cancer cells. In addition, the system could induce apoptosis by enhancing ROS production and disrupting the cytoskeleton. Lan et al. employed an inorganic carrier, made of galangin (Gal)-loaded mesoporous copper sulfide (CuS) NPs, in contrast to the study described above [124]. The GE11-CuS@Gal nanoplatform showed excellent tumor-targeting ability in HSC-3 tumor cells. High accumulation increased ROS levels and inhibited the Nrf/OH-1-mediated antioxidant pathway, leading to the inhibition of growth and migration of oral squamous cell carcinoma (OSCC).

Beyond small molecular drugs and proteins, nucleic acid-based therapeutics were also efficiently delivered by GE11-targeted nanocarriers. Supramolecular polymeric NPs assembled through host–guest interactions (cyclodextrin/adamantane) were post-functionalized with GE11 and the pH-sensitive fusogenic peptide GALA for tumor-targeted gene therapy using VEGF shRNA cargos [125]. The systemic delivery of GE11-GALA-conjugated NPs decreased intra-tumoral neovascularization and inhibited A549 tumor growth. Another study highlighted the delivery of mIRNA for gene silencing in T24 cells, based on core–shell mesoporous silica NPs (MSNPs) loaded with miR200c and surface modified with GE11-grafted amino-acid block copolymer to favor endosomal escape [126]. GE11 enhanced the EGFR-mediated uptake and large loaded MSNs (160 nm) exhibited a remarkable gene knockdown efficacy and antitumoral effect. Virus-like particles (VLPs) designed from *Macrobrachium rosenbergii* nodavirus (MrNV), which is a shrimp infectious and non-enveloped virus, were also conjugated to GE11 for the encapsulation of EGFP DNA plasmid [127]. The presence of the targeting peptide allowed for enhanced binding and internalization in EGFR-overexpressing colorectal cancer cells (SW480), through a receptor-specific internalization pathway.

The targeting peptide GE11 was associated with NP-based targeted anticancer therapies in a variety of tumors, including triple-negative breast cancer [128,129,130], breast cancer [131,132], prostate carcinoma [133], hepatic carcinoma [134,135], pancreatic ductal adenocarcinoma [136,137], lung adenocarcinoma [138,139,140], colon adenocarcinoma [141,142], epidermoid carcinoma [143], oral squamous cell carcinoma [144] and cervical cancer [135].

### 3.3. Anti-EGFR Whole Antibodies

Back to the pioneering studies of Hericourt and Richet in 1895 and of Ehrlich in 1913, Abs were already described as “magic bullets” to target cytotoxic compounds to specific regions of the body. Abs, also called immunoglobulins, consist of two heavy and two light chains, which join to form a Y-shaped protein. The two identical heavy and light chains are connected by disulfide bonds [145]. The light chains are made of one variable domain V_L_ and one constant domain C_L_, whereas heavy chains are formed by one variable domain V_H_ and three to four constant domain C_H_. Abs are large proteins of about 10–15 nm in size [146] and a molecular weight of ~150 kDa [147]. Structurally, they are formed by two antigen-binding fragments (Fabs), responsible for the recognition and binding to the target, and a fragment crystallizable region (Fc), in charge of the activation of the immune system and binding to cell receptors. Abs are produced by immune B cells when they come into contact with an antigen. mAbs are artificially produced through genetic engineering to be specific towards any kind of antigenic site. Their very high specificity and availability have made mAbs one of the most used classes of targeting agents for cancer. Representative anti-EGFR mAbs already on the market include cetuximab, panitumumab, nimotuzumab, matuzumab, necitumumab and amivantamab [148]. Besides their targeting abilities, mAbs are also known for inhibiting tumor cell proliferation or angiogenesis by binding specifically to cell surface receptors that are unique or overexpressed by tumor cells, such as EGFR [149]. For the purpose of this review, we will not describe the mechanisms of mAb-mediated cell death, but we will focus on their use as targeting ligands.

Among biological ligands, mAbs have the longest history with respect to targeting specific receptors. Their characteristic three-dimensional shape provides very high affinity for specific substrates. However, due to their dimension, the immobilization of mAbs to NP-based carriers results in significant increases in size and surface patterning, which may lead to immune responses. Moreover, their correct orientation must be preserved upon conjugation [150], while both random and oriented immobilization routes were reported [151]. Among the possible spatial orientations at NP surfaces, the “end on” (conjugation to Fc) topology is preferred for targeting applications, in order to ensure optimal exposure and accessibility of the Fab which is responsible for the antigen recognition.

The recombinant human/mouse chimeric mAb cetuximab (C225) was the first FDA-approved anti-EGFR mAb, characterized by a molecular weight of 145.8 kDa and a high binding affinity for EGFR (Kd = 0.201 nM). The use of C225 for active cancer targeting of nanotherapies was abundantly reported.

C225 functionalization has been extensively investigated on both organic and inorganic nanocarriers. In the first category, Fang et al. produced C225-modified lipid NPs for chemo-phototherapy of EGFR-overexpressing CRC [152]. The system, consisting of PLGA-lipid-based NPs, demonstrated enhanced pH/NIR-triggered drug release, photothermal response, cellular uptake and ROS generation compared with the non-targeted counterparts. Under NIR irradiation, the system exhibited IC_50_ values of 22.84 ± 1.11 μM at 24 h and 5.01 ± 1.06 μM at 48 h. Juan et al. developed C225-conjugated polylactide (PLA) NPs via polyethyleneimine (PEI) cross-linking (ACNPs) [153]. In vivo, the targeted system exhibited enhanced targeting of EGFR-expressing head and neck tumors in a xenograft model. Furthermore, alpelisib-loaded ACNPs significantly reduced cell viability and induced apoptosis. Duwa et al. conjugated C225 to temozolomide (TMZ)-loaded poly(lactic-co-glycolic acid) (PLGA) NPs (Cmab-TMZ-PLGA-NPs) [154]. In vitro, EGFR-overexpressing U-87MG cells showed higher TMZ uptake compared with SW480 or SK-Mel 28 cells, with the C225-conjugated system outperforming the non-targeted one. A similar polymeric system based on PLGA was exploited to conjugate C225 to docetaxel (DTX)-loaded NPs for the treatment of NSCLC [155]. The targeted NPs exhibited higher uptake in EGFR-overexpressing lung cancer cells A549 and NCI-H23. Furthermore, the targeted platform improved therapeutic efficiency with lower IC_50_ values, and significantly reduced tumor growth and proliferation in lung cancer mouse models. Hosseini et al. developed C225-functionalized poly amido amines (PAMAM) nanocarriers labelled with Lutetium-177 (^177^Lu) for theranostic applications [156]. In the EGFR-overexpressing SW480 cells, the targeted system showed enhanced uptake and significant anti-tumor effects. In vivo, the targeted nanoplatform exhibited rapid blood clearance and high tumor targeting, with minimal uptake in non-target organs. Yue et al. developed a C225-polymersome-mertansine nanodrug (C-P-DM1) for targeted therapy of EGFR-positive cancers [157]. The targeted system showed significantly higher cellular uptake in EGFR-overexpressing MD-MB-231, SMMC-7721 and A549 cells, with 10.5-, 35.8- and 18.2-fold increases, respectively. Furthermore, C-P-DM1 exhibited the greatest cytotoxicity in MDA-MB-231 and SMMC-7721 cells, with IC_50_ values of 33.8 and 27.1 nM, respectively, significantly lower than P-DM1. In vivo, C-P-Cy5 rapidly accumulated at the tumor site in MDA-MB-231 TNBC-bearing mice, without causing toxic side effects (Figure 6).

Another type of organic carrier that was investigated for C225 functionalization consists of albumin NPs. Ye et al. developed C225-modified albumin NPs loaded with MC-Val-Cit-PAB-DOX as antibody-drug conjugates (ADCs) for targeted therapy [158]. In vitro, the targeted system showed specific uptake in EGFR-overexpressing RKO cells, with minimal uptake in LS174T cells. Cytotoxicity assays confirmed the greater antitumor effect in RKO cells. In vivo, the targeted platform inhibited tumor growth in RKO-tumor bearing mice without causing significant weight loss, while free doxorubicin (DOX) caused severe toxicity and led to 20% mortality. Similarly, egg serum albumin NPs were functionalized with C225 and crosslinked with glutaraldehyde to target and treat Caco-2 colon cancer cells [159]. Cytotoxicity assays showed that the targeted formulation had the highest antitumor efficacy after 24 h (IC_50_ of 120 μg/mL), outperforming C225, albumin NPs and pure albumin. The enhanced toxicity and apoptosis were attributed to improved internalization and penetration via the targeted platform.

Combining organic and inorganic nanocarriers, Chen et al. integrated MSNPs and PEGylated lipid bilayers (SLBs) to achieve a C225-targeted hybrid nanoplatform (SLB-MSN) for the treatment of CRC [160]. In vitro, the targeted system showed enhanced toxicity against HCT-116 cells compared with free 5-FU, MSN/5-FU and SLB-MSN/5-FU, with higher efficacy linked to elevated EGFR expression. In vivo, biodistribution studies showed significantly higher accumulation in HCT-116 tumors than SW-620 tumors, confirming the tumor-specific targeting ability of the platform. Another example displaying the combination of organic and inorganic carriers is found in the work of Dorjsuren et al., who grafted C225 to the surface of thermo-sensitive liposomes (TSLs) and loaded them with iron oxide NPs and DOX [161]. In EGFR-overexpressing SKBR-3 cells, the targeted system showed higher uptake than non-targeted TSLs, while no significant difference was detected in EGFR-low expressing MCF-7 cells.

Several studies on C225 functionalization have also been reported on inorganic nanocarriers. Ma et al. developed C225-conjugated perfluorohexane (PFH)/AuNPs for low-intensity focused ultrasound (LIFUS) diagnosis ablation of anaplastic thyroid carcinoma (ATC) [162]. In vitro, the targeted platform triggered apoptosis in C625 thyroid carcinoma, while in vivo the system exhibited antitumor properties in human thyroid carcinoma xenografts. C225-functionalized AuNPs were also exploited for the treatment of metastatic CRC [163]. Among tested sizes, 60 nm C225-AuNPs exhibited the highest cytotoxicity in BRAF-mutant HT-29 cells, inducing 27.86% apoptosis compared with 15.68% for free C225, and 12.22% for free AuNPs. The enhanced efficacy was attributed to increased EGFR endocytosis and downstream signaling suppression. Wang et al. produced a C225-decorated drug delivery system based on NIR-activated NPs loaded with Pt(IV)-prodrug and indocyanine green (ICG) [164]. In EGFR-overexpressing A431 cells, the targeted system exhibited higher uptake and accumulation compared with minimal uptake obtained with free ICG, and NIR irradiation showed to enhance the efficacy even more. Moreover, the C225-decorated nanoplatform resulted in superior therapeutic effects and cell viability compared with free cisplatin, likely due to the sustained release of Pt(IV)-prodrug from the NPs.

Other anti-EGFR mAbs, including nimotuzumab (Nm) and panitumumab (Pm), were investigated for conjugation to nanocarriers.

Nm is an IgG1 mAb against human EGFR, with a lower immunogenicity profile than other mAbs [165]. Due to a lower affinity (Kd of ~20 nM) [166] and bivalent binding to EGFR, Nm usually targets cells with high EGFR density, resulting in reduced binding to normal cells and lower toxicity profiles than C225 [167]. Nm-AuNPs conjugates, assembled through thiol-mediated surface immobilization, achieved 4- to 5-fold enhanced cell growth inhibition compared with free Nm on the EGFR-overexpressing skin cancer cells A431, and EGFR-low expressing lung cancer cells A549 [168]. Nm also enhanced the uptake of AuNPs through specific EGFR-targeted activity.

Pm is an IgG2 isotype mAb, presenting almost 8-fold higher EGFR affinity than C225 (Kd of 0.05 nM) [169], and reduced immunogenic activity. Seminal reports on the use of Pm for targeted anticancer nanotherapies include the EGFR-mediated enhanced uptake of Pm-conjugated PLGA NPs loaded with TMZ, which resulted in increased ROS levels in EGFR-overexpressing U-87 MG cells [170]. Similarly, Pm-functionalized polycaprolactone (PCL) NPs improved the cytotoxicity of bosutinib in HCT-116 cells and induced 88% reduction in tumor size in mice with severe combined immunodeficiency disorders [171].

In the recent literature, full Abs were reported to improve the performance of NP-based carriers [172,173] for the delivery of small molecular chemotherapeutics, such as 5-fluorouracil [174,175], DOX [161,176], irinotecan [152,177], alpelisib [153] and DTX [178].

### 3.4. Anti-EGFR Antibody Fragments

A growing interest was observed over the past decades for the development of smaller Ab fragments, derived from conventional full Ab structures or produced recombinantly [179]. Despite their high binding affinity toward their target receptor [43], full Abs suffer from multiple limitations, including (i) bulky nature and high molecular weights; (ii) high immunogenicity; (iii) poor stability; and (iv) challenging control over oriented conjugation [180,181,182]. Ab fragments appeared as valuable substitutes due to their small size, which allows increased tumor penetration, their low immunogenicity due to the lack of Fc region, ease of production, high binding affinity, permeability across all biological barriers, high loading capacities and controlled orientation [183,184,185]. High tissue penetration is usually paired with faster renal clearance, a limitation that can be overcome with the conjugation of the targeting ligand to nanocarriers [186].

A meta-analysis by Mittelheisser et al. [187] of 161 studies (2009 to 2021) found that targeting efficiency of mAb-NPs or fragment mAb-NPs depends on the nanocarrier type. Fragment mAb-NPs exhibited higher tumor uptake with lipid NPs, while full mAb-NPs performed better with polymeric and organic/inorganic NPs. Smaller platform sizes consistently correlated with higher targeting efficiency. However, the underlying mechanism responsible for this phenomenon remains unidentified.

Fragmentation of full mAbs can be achieved via enzymatic digestion or recombinant expression [188,189]. Pepsin digestion yields bivalent F(ab’)2 fragments (110 kDa), while papain digestion produces monovalent Fab fragments (50 kDa). Fab’ fragments (55 kDa) result from the reduction of F(ab’)2 fragments, leaving a free sulfhydryl group. Smaller fragments, like Fv, retain antigen-binding properties but lack constant regions, while single chain variable fragments (scFv) link variable regions with a short peptide. Novel formulations include domain antibodies (dAb), engineered single variable domains, and diabodies, bivalent dimers with linked V_H_ and V_L_ domains [179,190].

Table 1 presents recent examples of studies exploring NPs modified with anti-EGFR Ab fragments for cancer active targeting.

Other recent reports highlighted the use of single Ab fragments [195,196,197,198], combinations of Ab fragments [199,200], or combinations of Ab fragments with full Abs [201] for the targeting of NP-based delivery platforms.

### 3.5. Anti-EGFR Nanobodies

Nanobodies (Nbs) are artificially designed antibody mimetics, representing the smallest intact antigen-binding fragment. Nbs are considered single domain Abs, since they consist of a single amino acid chain that forms only one domain [202,203], and their Kd is in the range of 2–25 nM [204]. These heavy-chain variable domains (V_HH_) are antigen-binding fragments of camelid and shark heavy-chain Abs (HcAbs) with dimensions in the nanometer range (4 nm long and 2.5 nm wide) [205]. They are characterized by a molecular weight of ~12–15 kDa (10 times smaller than conventional Abs and 2–4 times smaller than Fab fragments or scFv), which provides them with excellent tissue penetration properties. The small sizes are also accompanied by reduced interaction surface with the antigen-binding site, leading to the recognition of unique epitopes which are often hidden to full Abs. Furthermore, they possess good solubility due to their highly hydrophilic nature. Nbs are non-immunogenic since they lack constant domains, which are usually responsible for the activation of the complement system and antibody-dependent cell-mediated cytotoxicity (ADCC) due to the Fc region. Nbs are much more stable than conventional Abs, they can be stored at high temperatures and can be effectively renatured after thermal denaturation. Furthermore, they maintain their biological activity under strong acid and basic conditions. However, Nbs suffer from short half-lives and rapid clearance from circulation, which could limit their clinical applications. Nevertheless, their half-life can be extended by PEGylation [206] or fusion with antiserum albumin [207,208].

Liu et al. demonstrated the targeting ability of anti-EGFR Ega1 Nbs conjugated to micellar constructs loaded with meta-tetra(hydroxyphenyl)chlorin for selective PDT [209] (Figure 7). EGFR-mediated internalization was observed in A431 cells, resulting in enhanced cellular uptake and photocytotoxicity, as compared to control EGFR-low expressing HeLa cells. In vivo pharmacokinetics highlighted extended circulation of the micellar nanocarriers, as compared to the free drug.

Hernàndez et al. developed an anti-EGFR Nbs-targeted PDT platform (NiBh) for the selective treatment of oral squamous cell carcinoma (OSCC) in cats [210]. The targeted platform demonstrated high affinity for human and feline EGFR, effectively killing EGFR-overexpressing cells (LD_50_ in the nanomolar range), while leaving the surrounding EGFR-low expressing cells unharmed.

Other relevant studies have been conducted conjugating anti-EGFR Nbs to different types of nanocarriers for cancer active targeting [211,212,213,214].

### 3.6. Anti-EGFR Affibodies

Affibodies (Afbs) are scaffold-based affinity reagents [215], featuring the Z-domain, a designed variant of the IgG-binding protein A derived from *Straphylococcus aureus*. Afbs are composed of a three-helix bundle formed by 58 amino acids [216,217]. Variations in 13 of these amino acids give rise to a very large number of ligand libraries, from which powerful binders with high affinity and specificity can be isolated through a variety of display methods [218]. Phage display libraries provided receptor specific Afbs with affinities (Kd) in the μM to pM range [219], such as the matured anti-EGFR Afb Z_EGFR:1907_ which exhibits 2.8 nM affinity to EGFR [220]. As an alternative, Afbs can also be produced by solid phase peptide synthesis (SPPS), allowing site-specific incorporation of reactive moieties [221], avoiding time-consuming bacterial expression and protein purification. Afbs are made of a single polypeptide unit, which allows rapid folding, and they do not contain cysteine residues in their structure [222]. Born as a novel class of mAbs mimetics, Afbs possess several advantages over full Abs, including their smaller size (6 kDa vs. 150 kDa, respectively), which leads to faster and more effective tumor penetration and more rapid clearance. Moreover, the binding site and affinity are similar to those of Abs. They are highly soluble and stable in harsh conditions, including alkaline pH and elevated temperatures, that usually denature most proteins [223].

Afbs-functionalized nanocarriers have demonstrated significant versatility and have been investigated for a broad range of applications, including imaging, diagnostics, and therapeutic purposes. As an example of the first category, Wu et al. developed gadolinium (Gd)-encapsulated carbonaceous dots (Gd@C-dots) modified with Ac-Cys-Z_EGFR:1907_ (Gd@C-dots-Cys-Z_EGFR:1907_) as an MRI contrast agent, demonstrating specific EGFR targeting in vitro and in vivo [224]. Larger NP size (~20 nm) allowed the achievement of enhanced tumor affinity and targeting in EGFR-overexpressing xenografts, while maintaining efficient renal clearance. Smaller NP sizes (~3 nm), on the other hand, showed reduced signal enhancement due to poor receptor binding. As a tool for diagnostics, Afbs-functionalized polystyrene microbeads (AffiBeads) were designed and produced for the specific detection of EGFR-overexpressing exosomes as diagnostic markers for NSCLC [225]. The system allowed sensitive detection of as few as 12 exosomes per bead via flow cytometry. Among the several Afbs-based targeted formulations explored for therapeutic applications, anti-EGFR Afbs-modified porous platinum NPs (pPt NPs) were developed [226]. The system showed enhanced EGFR-specific targeting and tumor homing in A431 models, outperforming the unmodified platform. Moreover, the functionalized nanocarrier demonstrated superior radiotherapy sensitization, inhibiting HIF-1α expression and increasing DNA damage after only two treatments. Roy et al. developed an enzyme prodrug therapy based on upconverting NPs and photo-cross-linkable anti-EGFR Afbs (UC-ACD) [227], leading to a 4-fold higher retention in irradiated colorectal cancer cells in vitro (Figure 8). Upon binding, the targeted system converted 5-fluorocytosine to 5-fluorouracil, reducing tumor growth by 2-fold. In vivo, combination of UC-ACD and NIR irradiation increased tumor accumulation by 5-fold compared to controls.

Karyagina et al. formulated EGFR-targeting modular nanotransporters (MNT) using Z_1907_ Afbs as targeting ligand to deliver the cytotoxic agent ^111^In directly to tumor cell nuclei [228]. The use of the nanoplatform allowed the achievement of significantly higher cytotoxicity compared to free ^111^In, highlighting Afbs-MNT as a promising vehicle for targeted anticancer therapy.

Several other studies have been reported to investigate the effect of anti-EGFR Afbs-functionalized nanocarriers in vitro and in vivo [229,230,231,232].

### 3.7. Anti-EGFR Aptamers

Discovered 30 years ago, aptamers have been investigated as an alternative class of targeting agents in addition to peptides and proteins [233,234,235]. Aptamers are short single stranded DNA or RNA molecules of ~20–100 bps [236]. RNA aptamers have been shown to assume more diverse and intricate three-dimensional structures than DNA aptamers, allowing a higher number of conformations [237,238]. However, DNA aptamers are more stable than RNA aptamers, with in vivo half-life values in plasma of 30 to 60 min or a few seconds, respectively [239]. The use of native RNA aptamers is partly limited by their low stability; however, conjugation of RNA aptamers to NPs can shield the oligonucleotides from enzymatic degradation, thereby extending their in vivo half-life [240,241,242,243]. Additionally, various strategies have been explored to enhance RNA stability and mitigate nuclease degradation. These include chemical modifications, such as substitution of the natural hydroxyl group at the 2′ position of RNA bases with fluorine, protection of the 5′ and 3′ ends of RNA aptamers through capping, or circularization of RNA [244,245,246]. The ability of aptamers to fold into specific 3D conformations allows them to recognize and bind with very high affinity to their receptors, through electrostatic interactions, van der Waals forces and hydrogen bonds [247]. In the study of Li et al., one of the selected aptamers against hEGFR, E07, bound tightly to the wild-type receptor with a Kd of 2.4 nM [248].

Known as “chemical antibodies”, aptamers are chemically synthesized, which allows site-specific modifications of their nucleotide chains. One of the most famous processes of aptamer identification and synthesis is known as systematic evolution of ligands by exponential enrichment (SELEX) [235,249]. It consists of sequential steps, including incubation of random ssDNA or ssRNA libraries with the target molecules, separation of aptamer-target complexes from non-binding sequences, amplification of the target-bound nucleic acid sequences by PCR (for DNA aptamer selection) or RT-PCR and RNA transcription (for RNA aptamer selection), incubation of targets with a new library for the next round of enrichment and repeating for a fixed number of cycles. The repeated selection cycles allow an increase in the high affinity binding of the selected nucleic acids [250]. Advantages of the use of aptamers over Abs include smaller sizes (8–25 kDa), better tissue penetration, thermal stability, higher pH resistance, lower immunogenicity and toxicity in vivo, easier modification and conjugation, and easier synthesis and production with minimal batch-to-batch variations [251,252]. Furthermore, due to the SELEX identification process, aptamers can have an unlimited range of targets. However, there are still some areas in which Abs outperform aptamers. Regardless of their shelf stability, aptamers suffer from shorter half-lives in vivo due to renal filtration and nuclease digestion [253,254].

Ibarra et al. developed polymer NPs (CPNs) conjugated with either anti-EGFR or TNBC-specific RNA aptamers for phototherapy applications [255]. The anti-EGFR aptamer enhanced binding, cell internalization, and phototoxic effects in vitro compared to bare or scrambled aptamer-functionalized nanocarriers. In the development of therapeutic options for TNBC, Agnello et al. conjugated the anti-EGFR aptamer CL4 to (Cis-Pt)-loaded fluorescent polymeric NPs (PNPs-CL4) to achieve rapid and efficient uptake in EGFR-overexpressing MDA-MB-231 and BT-549 cells (10-fold higher uptake than the non-targeted NPs), increased toxicity (12-fold higher than the free drug), and enhanced intra-tumor accumulation in MDA-MB-231 tumor-bearing nude mice (Figure 9) [256]. Effective drug dosage was reduced by five in comparison with the minimum free Cis-Pt dosage.

To address the heterogeneity of cancer cells, anti-EGFR/CD44 dual-RNA aptamers were conjugated to solid lipid NPs functionalized with dexamethasone for DOX delivery to MDA-MB-468 cells, overexpressing EGFR and CD44 [257]. While monotargeted NPs induced a significant reduction of cell viability, the presence of dual targeting aptamers enhanced the cell death levels. In a similar strategy, Camorani et al. prepared a multifunctional platform displaying two different surface-conjugated RNA aptamers on iridium(III) (Ir_en_)-embedded Au-core/silica-shell-based NPs for synergistic PDT and PTT [258]. Anti-EGFR CL4 and anti-PDGFRβ RNA aptamers were selected for tumor and stromal cells targeting, respectively. The efficacy and synergistic effect of the dual-aptamer-decorated NPs were established on TNBC cells, luminal/HER2-positive breast cancer cells, epidermoid carcinoma cells, and adipose-derived mesenchymal stem cells. Tested on preclinical 3D stroma-rich breast cancer models, the multifunctional NPs were also effective in spheroids and organoids, highlighting the superiority of dual targeting.

The co-delivery of gefitinib and rapamycin, loaded in aptamer-conjugated chitosan NPs, was assessed to overcome EGFR-TKI resistance in NSCLC by promoting autophagy [259]. While enhanced cytotoxicity was observed in EGFR-overexpressing cells (H1975), non-targeted cell lines were not impacted. Furthermore, the high therapeutic efficacy was validated on tumor xenografts.

The targeting ability of aptamers was also combined with GSH-responsive polymeric NPs for the enhanced delivery of homoharringtonine to EGFR-overexpressing A549 breast cancer cells [260]. In vivo administration to A549 tumor-bearing nude mice confirmed the highest tumor-suppression potential of the targeted NPs.

Yang et al. developed anti-EGFR RNA aptamer-loaded NPs (EGFR_apt_-3WJ-si*KRAS^G12C^*) for targeted delivery of siRNA to NSCLC cells [261]. The platform showed enhanced cell association and KRAS suppression compared to the non-targeted nanocarriers, with higher uptake mediated by EGFR expression. In vivo, EGFR-targeted platforms led to higher tumor accumulation and efficient tumor suppression, unlike the non-targeted counterparts.

Other recent studies highlighted RNA aptamers-functionalized NPs for targeted breast cancer therapy [262,263], aptamer-conjugated nanoplatforms for the delivery of poorly water-soluble chemotherapeutic agent paclitaxel [262,264], aptamer-functionalized AuNPs [265,266], and nanocarriers functionalized with a combination of targeting aptamers and Abs [265].

Table 2 presents a summary of the reported anti-EGFR targeting ligands used for functionalization of nanocarriers.

## 4. Bioconjugation Strategies

There are several chemical strategies available to graft anti-EGFR ligands to the surface of nanocarriers, including covalent and non-covalent approaches. In this section, the various methodologies are described and significant examples of functionalization are discussed, highlighting the benefits and limitations of each strategy.

### 4.1. Covalent Conjugation Strategies

#### 4.1.1. Carbodiimide Chemistry

Amide bond formation stands among the most common conjugation strategies for NP functionalization. An amide bond is a stable linkage formed between a primary or secondary amine and a carboxylic acid. The reaction relies on the pre-activation of the carboxyl groups with the use of coupling agents, such as 1-ethyl-3-(-3-dimethylaminopropyl) carbodiimide (EDC), leading to an O-acylisourea intermediate [267]. N-hydroxysuccinimide (NHS) or its water-soluble version (sulfo-NHS) are often paired with EDC to improve the reaction efficiency, through the stabilization of the O-acylisourea intermediate [268].

Amide bond formation can be efficiently performed without spacing the reactive functionality (amine or carboxylic acid) from the NP surface, therefore maintaining the hydrodynamic sizes of the initial nanocarriers. The reaction is sensitive to pH and is more efficient when performed in acidic medium (pH 4–5) [267]. To prevent competition with extraneous carboxyl or amine groups, the reactions need to be run in buffers which are devoid of these reactive chemical moieties, such as 4-morpholinoethanesulfonic acid (MES) buffer or phosphate buffers.

This strategy is one of the most reported for NP functionalization because it is easy to perform and accessible. Furthermore, the amino and carboxyl chemical handles are naturally found in several biomolecules, such as proteins and peptides, where amino acid residues tend to be the target for conjugation. The common anchoring points include the ε-amino group of lysine side chains, the N-terminal primary amine, the carboxyl group on aspartic acid and glutamic acid, or the C-terminus [54]. While protein- or peptide-based targeting ligands do not require chemical modifications, nucleic acids are generally functionalized through site-specific reactions to introduce the amino or carboxyl moieties, allowing further amide bond coupling to the NP surface.

When applied to protein ligands such as Abs, this strategy does not allow for controlling their orientation due to the wide and random distribution of amino and carboxyl groups on their backbone. Surface immobilization through their antigen binding sites may reduce the specificity of the resulting nanoplatform [182]. However, a number of studies highlighted the use of amide-bond couplings for NP functionalization with anti-EGFR ligands.

Table 3 provides examples of carbodiimide chemistry used as a strategy for the conjugation of anti-EGFR targeting ligands to NPs.

Although this chemical strategy is one of the most-employed, accessible and easy to perform, it is characterized by low site specificity due to the generally high abundance of carboxyl and amino groups in biological ligands, such as Abs. This can therefore lead to a lack of ligand orientation, impacting the overall final targeting efficiency [271].

#### 4.1.2. Schiff Base Reaction

The derivatization of NP surfaces with aldehyde functionalities is used to immobilize targeting ligands containing primary amines through the formation of imines, also known as Schiff bases [272,273]. Similar to amide coupling, this technique does not allow for controlling the orientation of conjugated Abs or peptides. Alternatively, mild oxidation of the carbohydrate units of the Ab-Fc region leads to the introduction of aldehyde or ketone moieties that can be selectively reacted with hydrazine/hydrazide-modified NPs to control the orientation of the immobilized ligands through hydrazone bonds [274].

Following this pathway, Jordan et al. performed a directional conjugation of an anti-EGFR Ab to the surface of barium titanate NPs (BTNPs) [173]. The Ab was first reacted with sodium periodate to induce oxidation of the Fc region, yielding reactive aldehyde groups, which were covalently conjugated to a *N*-ε-maleimidocaproic acid hydrazide (EMCH) cross-linker via hydrazone bonds (Figure 10). The maleimide groups further reacted with thiol-modified BTNPs and the resulting thioesters ensured that the Ab binding region was facing outward to increase the targeting efficiency.

Similarly, Kawelah et al. reported a bifunctional PEG linker presenting amino- and DBCO-end functionalities for the stepwise conjugation to oxidized anti-EGFR Abs (imine formation) and azido-modified indocyanine green J-loaded polymersomes (copper-free click reaction) [275].

Another study highlighted the modification of iron oxide NPs with oxidized sodium alginate, allowing post-conjugation to the targeting peptide GE11 through imine formation [276]. The resulting nanocarriers were used for the MRI of nasopharyngeal carcinoma and targeted delivery of cisplatin.

Despite being less common than amide-based conjugations, Schiff base reactions favor proper spatial orientation of Ab ligands, resulting in enhanced targeting capabilities [277].

#### 4.1.3. Thiol-Maleimide Chemistry

Another very common conjugation strategy for NPs functionalization with targeting ligands involves the 1,4-addition of thiols to maleimide groups, which can be performed at neutral pHs and under mild conditions. This strategy requires the presence of free sulfhydryl groups, either on the ligands or on the NPs surface, which can be achieved in two main ways: (i) reduction of native disulfide bonds in the presence of mild reagents, such as tris(carboxylethyl)phospine (TCEP), dithiothreitol (DTT) or 2-mercaptoethanol (BME) [278]; or (ii) conversion of amines into thiols by using 2-iminothiolane (Traut’s reagent) and N-succinimidyl S-acetylthioacetate (SATA) [279]. NPs are generally modified with maleimide groups, using heterobifunctional linkers. This strategy is more site-selective than carbodiimide chemistry due to the limited number of sulfhydryl groups in proteins, in comparison with amines, and benefits from faster kinetics at neutral pH [280]. However, thiol-maleimide chemistry does not allow proper control over the orientation of Abs-conjugated NPs.

Aptamers, Afbs and Nbs are subjected to site-specific modification with a free thiol group, allowing for selective thiol-maleimide conjugation. However, the resulting thioether linkages are sensitive to reducing potential and can be cleaved by exogenous glutathione (GSH), a reducing agent found naturally in the circulation and cellular compartments [281]. In addition, when using Ab-based targeting ligands, the reduction step might damage the Ab tertiary structure and decrease its binding activity [282].

Nevertheless, thiol-maleimide conjugation was used for the immobilization of a large variety of targeting ligands to nanocarriers. For example, Guo et al. successfully conjugated a free thiol-bearing GE11 peptide to a dual drug-loaded polymeric micellar system using a Michael-type thiol-ene reaction [116]. The approach ensured effective reaction of the terminal maleimide groups with GE11, leading to functionalized NPs based on amphiphilic polycarbonate, whose systemic administration in vivo resulted in increased tumor growth suppression and reduced metastasis and toxicity compared to the non-targeted platform. Mesquita et al. conjugated anti-EGFR Nbs to zinc phthalocyanine-loaded polymeric micelles via thiol-maleimide chemistry (Figure 11) [211]. The presence of the targeting ligand at the surface of the NPs generally enhanced the association and phototoxicity in cells overexpressing the target receptor.

Geddie et al. engineered Fabs with strategically placed disulfide bonds to avoid over-reduction during initial processing while preserving their binding ability [283]. When functionalized to immunoliposomes via thiol-maleimide chemistry, the engineered Fabs enhanced uptake in multiple cell lines compared to the non-targeted counterparts. Ye et al. designed ADCs by grafting a previously reduced C225 to DOX using a maleimide-bearing linker, followed by adsorption onto the surface of BSA NPs [158]. Flow cytometry analysis demonstrated higher binding and accumulation in EGFR-overexpressing cells. Furthermore, in vivo studies showed higher tumor inhibition efficacy and lower system toxicity for the targeted system compared to the non-targeted control. In another study, anti-EGFR Afbs were grafted to DOX-loaded PEGylated liposomes using thiol-maleimide chemistry [231]. In vitro studies revealed a higher DOX uptake in EGFR-overexpressing cells compared to EGFR-low expressing cells, highlighting the selective enhanced cytotoxicity. In vivo, the targeted platform exhibited long circulation time and efficient accumulation in EGFR-overexpressing tumors. Moreover, low-dose treatment could not only produce an ideal antitumor effect but also reduce its systemic toxicity.

#### 4.1.4. Dative Chemistry (Thiol-Metal Bond)

Another strategy for the direct conjugation of biomolecules to NP surfaces involves dative (coordination) bonds, which are less robust than covalent linkages, due to extended lengths, higher energy and sensitivity to pH variation, oxidation or replacement by similar molecules [284,285]. However, dative bonds can be enhanced by increasing the number of interactions. Known examples of dative bonds include chelation of metal ions (coordination by electron-donating amino acids) and gold-thiol chemisorption, which is commonly used for the functionalization of AuNPs with thiolated biomolecules, including proteins, peptides and oligonucleotides.

Table 4 presents examples of nanocarriers decorated with anti-EGFR targeting ligands via gold-thiol chemisorption.

#### 4.1.5. Click Chemistry

Click chemistry refers to a group of chemical reactions, described by Sharpless et al. in 2001 and presenting high potential for NP functionalization due to their high selectivity, favorable reaction rates, compatibility with mild aqueous conditions, minimal production of by-products and high yields [288,289]. Click reactions mostly include copper-catalyzed azide-alkyne 1,3 dipolar cycloaddition (CuAAC) [290], strain-promoted azide-alkyne cycloaddition (SPAAC) [291], and inverse electron-demand Diels-Alder reaction (IEDDA) between tetrazine (Tz) and trans-cyclooctene (TCO) [292,293]. Such reactions are considered “biorthogonal”, because the reactive groups are highly selective and inert toward other functionalities present in biological systems, such as carboxyls, amines, hydroxyls, thiols, alkenes, amides, esters, disulfides and phosphodiesters.

Despite this favorable profile, click reactions require the introduction of the non-native reactive handles on the targeting ligand and NP surface. For example, azido groups can be site-specifically introduced on the heavy chain of the Fc of Abs by using an enzymatic modification, leading to further controlled oriented conjugation (Figure 12) [294,295]. As a result, the conjugation of Abs to NPs was 5- to 8-fold more efficient through click reactions than amide reactions [296,297].

Both inorganic and organic NPs were efficiently functionalized through click reactions. Porous silicon NPs, modified with DBCO-terminated PAMAM dendrimers, were conjugated to azido-containing anti-EGFR Nbs for the co-delivery of siRNA and DOX [298]. Similarly, DBCO-modified UCNPs underwent SPAAC in the presence of photo-cross-linkable anti-EGFR Afbs-enzymes. The introduction of the azido reactive handle on the targeting construct involved a PEG spacer [227]. Alternatively, DBCO-modified anti-EGFR Abs were efficiently immobilized on azido-functionalized MSNPs [299].

The versatility of SPAAC reactions was further demonstrated by Tran et al. in the preparation of customized antisense oligonucleotides (ASOs)-loaded red blood cell-derived extracellular vesicles (RBCEVs) to selectively target mutations in the EGFR gene [300]. Their specificity was enhanced following SPAAC immobilization of azido-modified anti-EGFR Nbs. Other examples include the conjugation of C225 to azido-decorated polymersomes [157]. In these approaches, the introduction of the reactive handles, via bifunctional linkers, made use of conventional carbodiimide chemistry, therefore resulting in random orientation of the immobilized targeting ligands [301].

The demonstration of the oriented conjugation of anti-EGFR targeting ligands mediated by click chemistry was not yet reported. However, examples are found with similar receptors, such as CD11 [294] and HER2 [296].

### 4.2. Non-Covalent Conjugation Strategies

#### 4.2.1. Interaction by Adapter Molecules

One of the most represented non-covalent approaches exploits the strong binding affinity between biotin and biotin-binding proteins, such as avidin or its analogues, including neutravidin and streptavidin. Avidin is a tetrameric glycoprotein characterized by a molecular weight of ~68 kDa, and composed of four identical subunits of 128 amino acids. Avidin has been extensively reported to bind biotin with high affinity and specificity (Kd ~ 10^−15^ M), making the avidin-biotin interaction one of the most specific and stable non-covalent interactions [302,303]. NP surfaces can be coated with avidin (by electrostatic interactions or covalent conjugation strategies) and targeting ligands can be chemically modified with biotin for subsequent specific immobilization through avidin-biotin interactions. In particular, this technique offers an oriented conjugation of Abs to NP surfaces, since Abs can be engineered with biotin molecules at their Fc region. The main advantages of this strategy rely on its high stability and robustness against temperatures, pH, harsh organic solvents and other denaturing reagents [304]. However, avidin has a high degree of non-specific binding, due to its basic isoelectric point (pI) and high glycosylation degree. Superior variants of avidin include the deglycosylated streptavidin (pI of ~5–6) [305] and neutravidin (pI of ~6.3) [306,307].

Among the applications of this non-covalent approach, one can cite the functionalization of neutravidin-coated gold nanorods with biotinylated anti-EGFR Ab (Figure 13) [308], the production of C225-conjugated vaterite NPs for the release of immunotherapeutic proteins [309], and the decoration of red blood cell-derived extracellular vesicles with anti-EGFR Ab for the delivery of therapeutic RNA via intrapulmonary administration [310].

#### 4.2.2. Electrostatic Interaction

Non-covalent bioconjugation techniques rely on the adsorption of ligands to the surface of nanocarriers through electrostatic interactions, hydrogen bonding, hydrophobic interactions and van der Waals forces. Despite their cost-effectiveness and ease of implementation, non-covalent interactions are much less robust than covalent bonds and more prone to degradation. Furthermore, these interactions are non-specific and sensitive to changes in experimental conditions, such as temperature, ionic strength and pH. Since this type of chemistry often requires only stoichiometric mixing of the NPs and biomolecules, it can be referred to as “self-assembly”. Electrostatic adsorption approaches rely on the attraction between oppositely charged species: charged nanocarriers will attract oppositely charged targeting ligands [311]. For example, the strong negative charge of phosphate groups in oligonucleotides causes adsorption on the surface of positively charged NPs [312]. Proteins usually present a higher tendency to electrostatically adsorb to NP surfaces, with a trend that depends on their pI values. Research conducted on polystyrene NPs showed that proteins with lower pI (<5.5), such as albumin, mainly adsorb on positively charged NPs, whereas those with higher pI (~7–8), such as IgG, tend to adsorb on negatively charged NPs [313]. In general, neutral NPs are less prone to non-covalent immobilization of proteins [314]. The layer-by-layer adsorption technique relies on the sequential deposition of alternating layers of anionic and cationic polyelectrolytes on a substrate, providing high loading of ligands, since multilayer structures are formed on the surface of the nanocarriers [315,316,317,318].

Examples of electrostatic interactions used as a bioconjugation strategy include the synthesis of polyarginine-tailed anti-EGFR Afbs and its incorporation into methotrexate-loaded 11-mercaptoundecanoic acid-modified gold nanoclusters via charge-based self-assembly, forming a shell to seal in the loaded drug [319]. The targeted system exhibited good biocompatibility, efficient drug loading, precise targeting of EGFR-overexpressing cells and dual-responsive drug release. C225 was electrostatically adsorbed onto polydopamine NPs co-loaded with 5-fluorouracil, irinotecan and leucovorin for metastatic colorectal cancer therapy [320]. Cellular uptake studies performed in HTC116 and HT29 human cell lines revealed rapid NP internalization, beginning after 30 min of incubation. In vitro cytotoxicity assays showed a synergistic effect among the nanocarrier, the encapsulated drugs and the adsorbed mAb, resulting in a reduction in the survival rates of HCT116 and HT29 cells by 22% and 30%, respectively, after 72 h incubation. Liu et al. developed a multifunctional nanodrug delivery system based on gefitinib- and EGFR siRNA-loaded imidazolate framework-8 (Apt/(siRNA + GEF)@ZIF-8 NPs) to suppress the drug-resistant gene expression in tumors (Figure 14) [321]. Anti-EGFR aptamers were conjugated at the surface via electrostatic adsorption, allowing for enhanced accumulation at tumor sites.

Salama et al. investigated the decoration of the surface of PEGylated TiO_2_ NPs with EGF via physical adsorption processes [101], demonstrating the safety and increased cellular uptake of the platform when used on EGFR-overexpressing cells.

Despite its accessibility and ease of conjugation, the immobilization of targeting ligands through non-covalent interactions suffers from several shortcomings, including limited stability and random orientation of the surface-adsorbed ligands, which impacts their functionality. Anaki et al. investigated the targeting and therapeutic potential of C225, immobilized on AuNPs through covalent (amide bond) and non-covalent (adsorption) interactions [322]. First, the conjugation efficiency was enhanced at low initial Ab mass through adsorption and at high initial Ab mass through amide coupling. Then, despite the random orientation of surface-conjugated C225, the covalent strategy resulted in enhanced anti-cancer activity on human squamous carcinoma A431 cells, as compared to the non-covalent constructs.

#### 4.2.3. Fc-Binding Receptors Mediated Conjugation

As described above, most of the non-covalent conjugation strategies do not provide a controlled and oriented ligand conjugation, leading to reduced targeting efficiency. Alternatively, the use of Fc-binding protein spacers (i.e., protein A, protein G, protein A/G and Fc receptors) was developed to ensure oriented conjugation.

Hirata et al. reported the use of genetically modified staphylococcal protein A, containing a lysine cluster, for the sequential electrostatic immobilization on anionic liposomes and subsequent binding to the Fc region of anti-EGFR Abs [323]. The Anchored Secondary scFv Enabling Targeting (ASSET) technology [324], which relies on a membrane anchored protein, was applied to the oriented and uniform functionalization of LNPs with anti-EGFR Abs for cell-specific siRNA delivery, resulting in 100% bioconjugation efficiency [325].

Table 5 presents an overview of the available bioconjugation strategies of anti-EGFR ligands to nanocarriers, highlighting their advantages and limitations.

## 5. Characterization Techniques

The last section of this review highlights the main characterization methods (qualitative and/or quantitative) which are currently available to monitor the conjugation of anti-EGFR targeting ligands at the surface of nanocarriers.

### 5.1. Dynamic Light Scattering

Dynamic light scattering (DLS) is a common technique to evaluate the hydrodynamic size and surface charge (zeta potential) of nanocarriers, based on the intensity of light scattered by the particles during their movement in a medium (Brownian motion) [326]. DLS provides valuable insight into the surface and colloidal properties at different steps of functionalization. While NP size values have direct impact on their circulation time in the bloodstream and their ability for cell and tissue penetration [327], zeta potential values are linked to their stability in the studied medium. Ab conjugation to the surface of nanocarriers tends to increase their hydrodynamic size (up to 50 nm size increase), as a result of both the steric hindrance of the immobilized ligand [328] and potential NP aggregation [329]. The variation of zeta potential upon functionalization is less significant and depends on several parameters, including the initial NP surface charge (nature and density of reactive moieties), the pI value of the ligand and the pH of the medium (Figure 15) [256,258]. The conjugation of smaller targeting ligands, such as aptamers, leads to smaller variations of the NP hydrodynamic size. For example, Li et al. observed a 3 nm increase of the size of branched RNA four-way junction NPs upon functionalization with EGFR-specific aptamers [330]. In such cases, the assessment of surface conjugation by means of DLS measurement is less reliable.

Noticeably, the hydrodynamic size of nanocarriers might also be reduced upon surface functionalization. Cruz de Sousa et al. reported that cabazitaxel-loaded liposomes evolved from 136.7 ± 2.2 to 95.0 ± 3.9 nm upon thiol-maleimide mediated conjugation of C225 [331]. The authors validated the Ab immobilization by gel electrophoresis and ELISA test, which confirmed the preserved structure and conformation for receptor binding.

DLS measurements provide valuable indications on the evolution of the hydrodynamic sizes and surface charge of nanocarriers along functionalization pathways. However, additional characterizations are needed to give evidence for the conjugation to targeting ligands.

### 5.2. Fluorescence Experiments

A more reliable and quantitative characterization method consists in the fluorescence detection of targeting agents, mainly Abs, at the NPs surface, using either the direct fluorescence emission of labelled targeting Abs or the indirect fluorescence emission of a labelled secondary Ab. Recording the fluorescence emission levels of Alexa Fluor 555-labelled anti-EGFR Abs conjugated to BTNPs allowed to quantify the Ab concentration in studied samples, with values of 4.59 ± 1.67 μg/mL [173]. For the indirect characterization of anti-EGFR Ab conjugation to PLGA-based NPs, secondary complementary Abs labelled with Dylight 488 [332] or fluorescein (Figure 16) [333] were reported.

### 5.3. Electrophoretic Techniques

Surface bound proteins and oligonucleotides can be evidenced by electrophoretic techniques. In particular, proteins are separated by molecular mass using sodium dodecyl sulfate (SDS)-polyacrylamide (PAGE) gel electrophoresis [334,335], while agarose gel electrophoresis is applied to charged molecules like DNA/RNA [336]. Successful functionalization is indicated by a fluorescent band of ligand-functionalized NPs at the baseline, with no visible band for free ligands (Figure 17).

Table 6 presents examples of nanocarriers functionalized with anti-EGFR targeting ligands and characterized through electrophoretic techniques.

Similar to light scattering methods, electrophoretic techniques do not provide a precise quantification of the grafted agents, but can be used as qualitative assessment of the presence of the ligands.

### 5.4. Protein-Based Assays

When protein-based targeting ligands are used, two commonly employed colorimetric quantification methods are the Bradford assay [340] and the bicinchoninic (BCA) assay [341].

The first one is based on the absorption shift—from 465 to 595 nm—that is observed in the spectrum of the Coomassie brilliant blue dye when bound to a protein [342]. This simple method allows for quantification of surface-conjugated proteins and is compatible with most solvents, reducing agents and salts. However, it cannot be performed in the presence of surfactants, even at low concentrations, and suffers from high protein-to-protein variation and linearity over a narrow window of concentrations.

More sensitive and reliable than the Bradford assay is the BCA assay. The BCA assay relies on the formation of Cu^2+^-protein complex under alkaline conditions and subsequent reduction to Cu^+^ ions, which are quantified upon formation of a Cu^+1^-(BCA)_2_ complex characterized by a deep blue color and absorption maximum at 562 nm [343]. This assay is characterized by a detection range of 20–2000 μg/mL (5–250 μg/mL for enhanced BCA assay) and is less prone to interference from non-protein sources, including lipids and detergents, that can impact other protein-based assays. In addition, it provides a high protein-to-protein uniformity, not being affected by differences in protein compositions. However, BCA assay is longer to perform, with incubation times ranging from 30 min to 2 h. Furthermore, the presence of reducing agents or copper chelating agents could impact the reliability of the results.

Table 7 shows examples of nanocarriers functionalized with peptides or proteins anti-EGFR targeting ligands and characterized through protein-based techniques.

In conclusion, the presence and density of protein ligands conjugated to nanocarriers can be efficiently assessed by the BCA and Bradford assays.

### 5.5. Spectroscopy Techniques

Spectroscopy methods, such as Fourier Transform IR spectroscopy (FTIR), nuclear magnetic resonance (NMR) and X-ray photoelectron spectroscopy (XPS), can be used for qualitative investigation of NP surface-conjugated targeting ligands. FTIR allows for the identification of surface functionalities according to their vibrational signatures (Figure 18) [348,349], and for their quantification in case a standard curve of known concentrations of the sample is available. NMR is a non-destructive physicochemical analytical technique capable of elucidating structural and dynamic properties of complex nanocarriers [350]. While detailed characterization at the atomic level is not possible for high molecular mass systems beyond 50 kDa, solid- and liquid-state NMR provide insights into the chemical structure and location of surface bound ligands and the nature of their interactions with the nanocarrier. Quantitative or semi-quantitative evaluation of the density of surface functionalities can also be achieved by NMR [351]. XPS is extensively used for surface characterization, providing semi-quantitative analysis of the chemical elements composing the material surface, except hydrogen and helium [352]. Applied to functionalized NPs, XPS allows to assess surface elemental composition, presence of contaminants, density of surface immobilized molecules, and thickness of layers and coatings [353].

The examples listed in Table 8 illustrate the versatility of spectroscopy techniques to investigate functionalized nanocarriers and characterize their payloads and surface-conjugated targeting ligands.

### 5.6. Thermogravimetric Analysis

Thermogravimetric analysis (TGA), which evaluates the change in mass of a sample as a function of temperature and time under controlled conditions, can be applied to the detection of organic ligands conjugated to inorganic NPs [355]. While it does not require any special sample preparation beyond drying, TGA is a destructive characterization method, consuming several milligrams of material [356].

Wang et al. developed neutron-activated ^153^Sm-filled multi-walled carbon nanotubes, which were further conjugated to C225 and evaluated for their therapeutic efficiency in a lung metastatic melanoma model [357]. TGA analysis of the constructs at 650 °C allowed for the quantification of immobilized Ab (220 mg/g material, Figure 19), in line with the value obtained via the BCA assay (173 mg/g).

However, another study focusing on EGF-functionalized nanoceria pointed toward a diminution of the weight loss after ligand conjugation, which was probably resulting from partial detachment of the coating layer upon EGF immobilization [358].

TGA is therefore a valuable technique to assess and quantify immobilized ligands at the surface of nanocarriers but is limited to the study of functionalized inorganic NPs.

Table 9 provides a summary of the available characterization methods to monitor the correct grafting of anti-EGFR targeting agents to nanocarriers.

## 6. Conclusions

Active targeting has been shown to offer several advantages over passive targeting in improving the efficacy and specificity of cancer therapeutics. This type of approach relies on the presence of ligands that recognize specific receptors, including EGFR, which are overexpressed on the surface of cancer cells. Nanodelivery systems functionalized with EGFR-targeting ligands at their surface have been increasingly investigated as promising tools to achieve enhanced specificity via receptor-mediated endocytosis.

Several anti-EGFR targeting agents, such as peptides, proteins, Abs, Ab fragments and oligonucleotides, have been identified and are currently used. The selection of the most appropriate ligand for NP functionalization depends on the specific application, since they all have different binding affinities for EGFR and each class of ligands has its own benefits and drawbacks. Various bioconjugation strategies are available for grafting anti-EGFR agents to nanocarriers’ surfaces, among which are non-covalent and covalent linkages. The choice of the conjugation chemistry plays a vital role in the determination of the final stability and efficacy of the functionalized nanocarrier. In general, opting for a chemical approach that allows for achieving oriented conjugation of the ligands leads to improved interaction between the targeting agents and the receptors, resulting in enhanced targeting efficiency. In addition, reliable characterization techniques are essential to investigate the surface functionalization pattern, including the chemical composition, integrity and density of immobilized ligands, the nature of the conjugation interactions and the stability of the nanoconstructs.

In conclusion, nanocarriers modified with anti-EGFR targeting ligands represent a promising approach for targeted cancer therapy, offering enhanced specificity and efficacy for cancer treatment. The integration of the design of functionalized nanocarriers, appropriate conjugation strategies and reliable characterization methods is critical to improve cancer therapy and patient outcomes.

## Figures and Tables

**Figure 1 nanomaterials-15-00158-f001:**
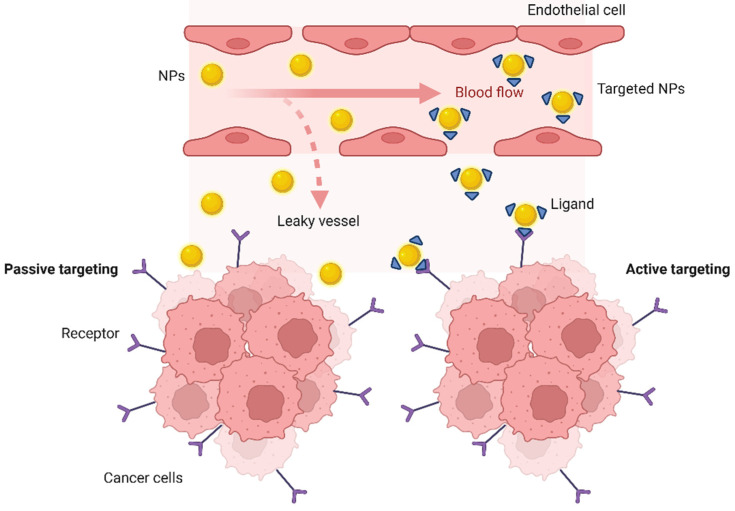
Schematic representation of active targeting vs. passive targeting of cancer cells by nano-delivery systems. In passive targeting, the size and surface properties of nanocarriers facilitate their accumulation within tumor tissues through the EPR effect, resulting from the leaky vasculature of tumors. Conversely, active targeting involves the conjugation of specific ligands to the NP surface, enabling targeted delivery to cancer cells by binding to overexpressed receptors or markers on their surface. Created with BioRender.com.

**Figure 2 nanomaterials-15-00158-f002:**
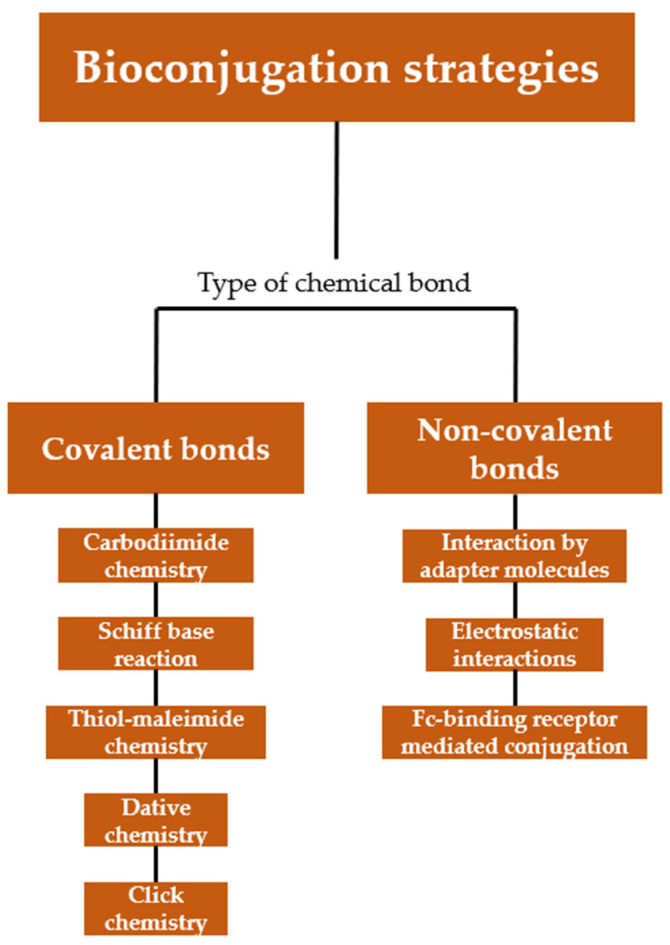
Classification of bioconjugation strategies based on covalent and non-covalent strategies.

**Figure 3 nanomaterials-15-00158-f003:**
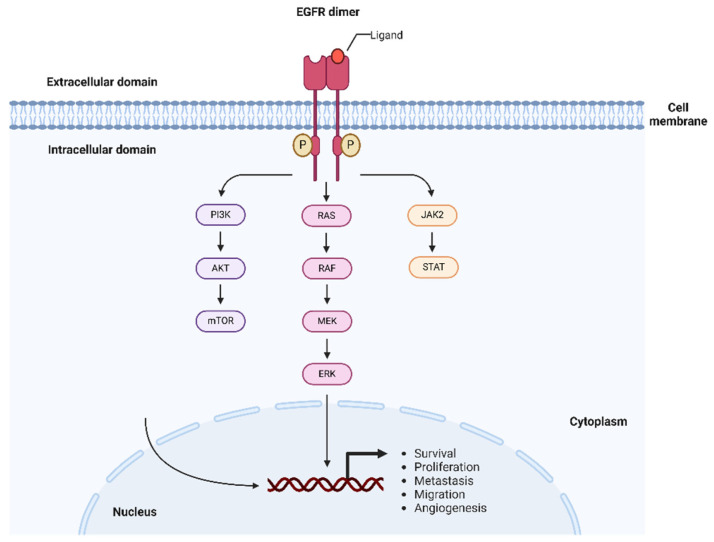
An overview of the EGFR signaling pathway. Upon binding of specific ligands, EGFR undergoes dimerization and phosphorylation, leading to the activation of downstream signaling cascades and resulting in enhanced tumor growth, invasion and metastasis. Some of the main activated pathways include PI3K/AKT, RAS/MAPK, and JAK/STAT, which cross-regulate and interact, contributing to cell proliferation, migration and survival. Created with BioRender.com.

**Figure 4 nanomaterials-15-00158-f004:**
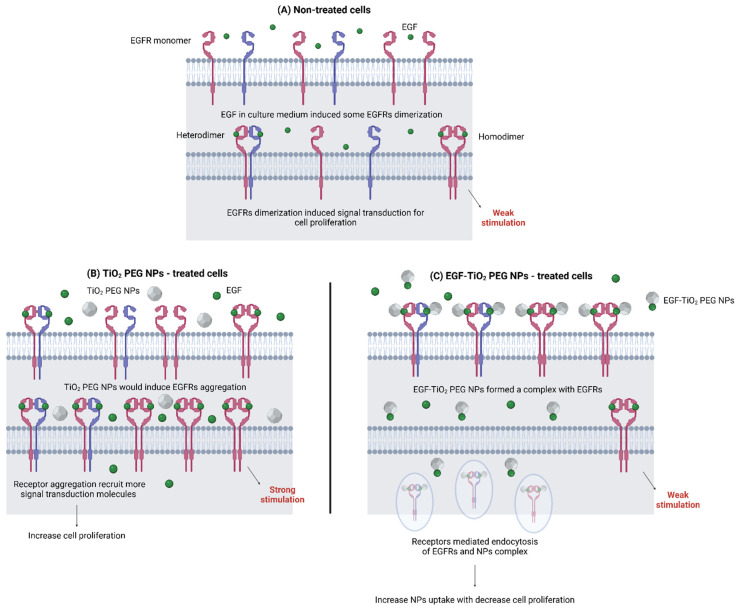
Molecular mechanism for the effect of EGF conjugation on TiO_2_ PEG NPs uptake levels and the cell proliferation effect via interaction with EGFRs proposed in [101]. (**A**) Non-treated A431 cells; (**B**) TiO_2_ PEG NPs-treated A431 cells; (**C**) EGF-TiO_2_ PEG NPs-treated cells. Created with BioRender.com.

**Figure 5 nanomaterials-15-00158-f005:**
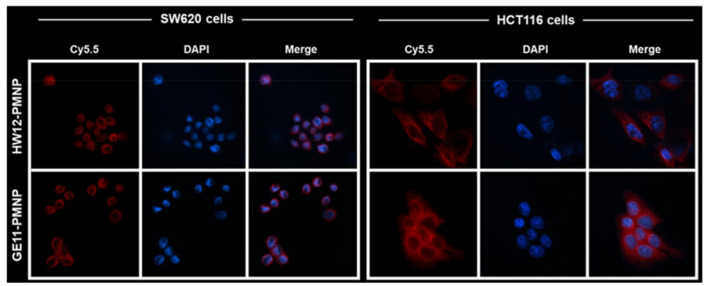
Confocal microscopy images of bare PMNPs and GE11-PMNPs in red (Cy5.5), nucleus in blue (DAPI), and their combination (merged) in SW620 and HCT116 cells. Adapted with permission from [117]. Copyright 2020, American Chemical Society.

**Figure 6 nanomaterials-15-00158-f006:**
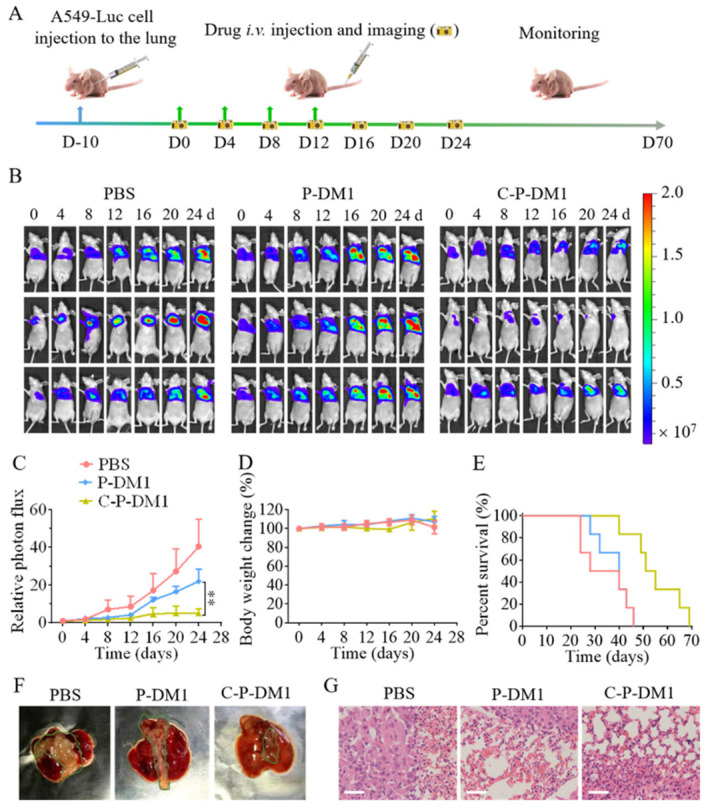
Therapeutic efficacy of C-P-DM1 in orthotopic A549-Luc NSCLC-bearing mice. (**A**) Establishment, therapy and monitoring scheme of the orthotopic A549-Luc NSCLC model. (**B**) Bioluminescence imaging and (**C**) quantitative bioluminescence levels of mice from day 0 to day 24 (*n* = 3). (**D**) Body weight change and (**E**) survival curves of mice (*n* = 6, C-P-DM1 vs. P-DM1 and PBS, ** *p* < 0.01). (**F**) Photographs and (**G**) H&E stained images of lung isolated from different groups on day 24. Scale bars are 50 μm. Adapted with permission from [157]. Copyright 2021, American Chemical Society.

**Figure 7 nanomaterials-15-00158-f007:**
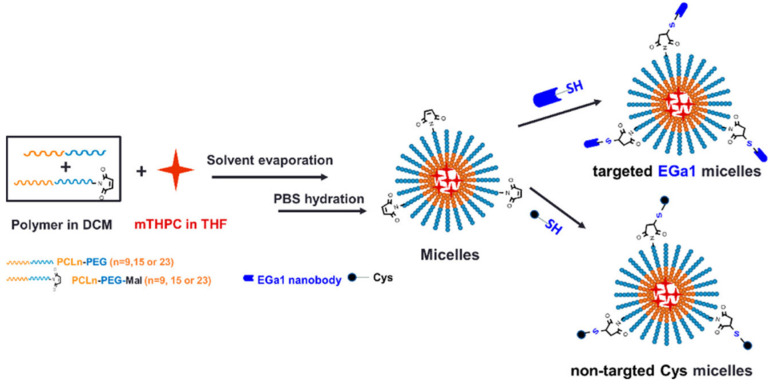
Preparation of polymeric micelles conjugated with EGa1 (targeted) or Cys (non-targeted). Adapted with permission from [209]. Copyright 2020, American Chemical Society.

**Figure 8 nanomaterials-15-00158-f008:**
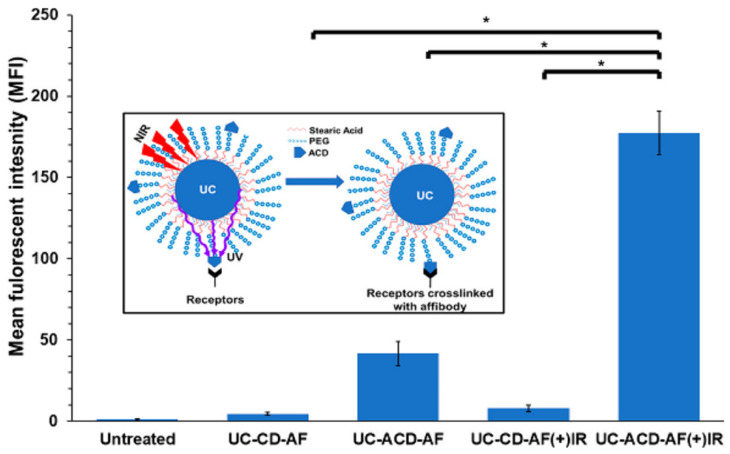
Mean fluorescent intensity from human colorectal Caco-2 cancer cells after incubation with Alexa Fluor 555-modified UC-ACD (targeted) and UC-CD (non-targeted), with and without NIR irradiation. MFI of UCNP-protein complex is 4.38 ± 0.32 times higher in the cells treated with UC-ACD-AF (+) IR than that of the non-irradiated control (UC-ACD-AF, *n* = 3, * *p* < 0.05). Inset: schematic showing the UC-ACD conjugate structure and the NIR activation of the UCNPs, followed by upconversion to UV, resulting in a photoreaction between the affibody-enzyme ACD and EGF receptors. This chemical association allows enzymatic conversion of systematically delivered prodrug 5-FC to active drug 5-FU at the cancer site. Adapted with permission from [227]. Copyright 2022, American Chemical Society.

**Figure 9 nanomaterials-15-00158-f009:**
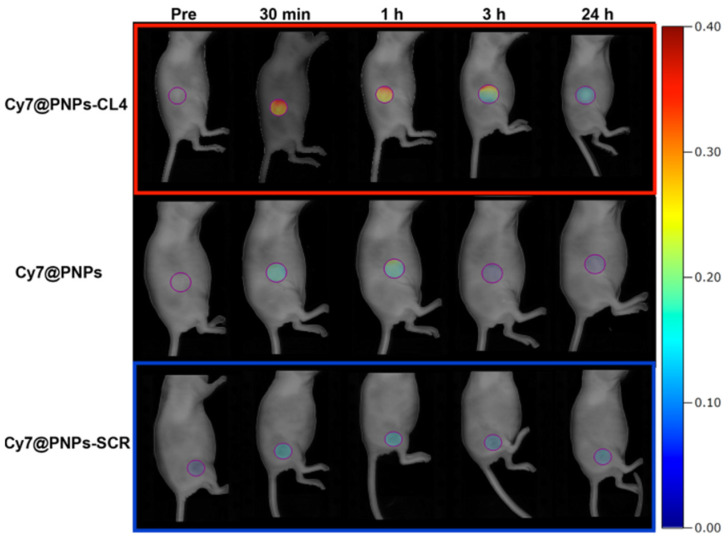
Selective tumor targeting of Cy7@PNPs-CL4 compared to Cy7@PNPs-SCR and Cy7@PNPs. Nude mice bearing subcutaneous MDA-MB-231 xenografts were i.v. injected with Cy7@PNPs-CL4, Cy7@PNPs or Cy7@PNPs-SCR (5 nmol Cy7/100 μL) and analyzed by in vivo FRI imaging at the indicated time points (i.e., Pre: before injection, 30 min, 1, 3 and 24 h acquisitions). Adapted from [256]. Copyright 2021, Springer Nature.

**Figure 10 nanomaterials-15-00158-f010:**
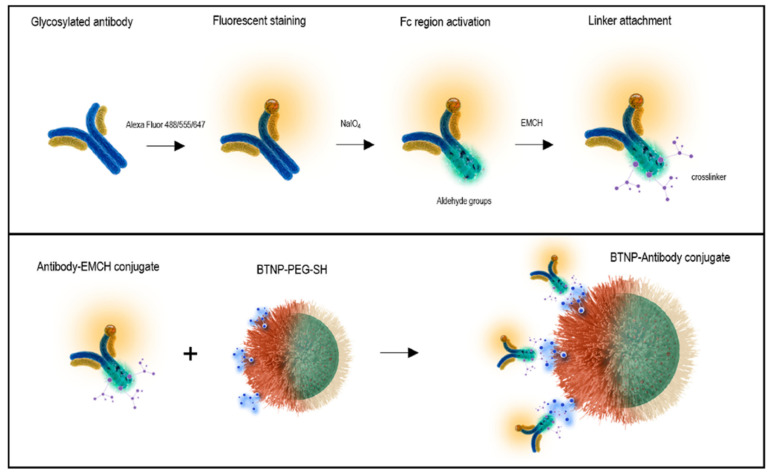
Schematic representation of the directional Ab conjugation chemistry. A glycosylated Ab is first fluorescently labelled, and then aldehyde groups are created on the Fc region. The hydrazide portion of an EMCH cross-linker binds to the aldehyde groups while the maleimide portion attaches to the thiolated BTNP surface. Adapted with permission from [173]. Copyright 2020, American Chemical Society.

**Figure 11 nanomaterials-15-00158-f011:**
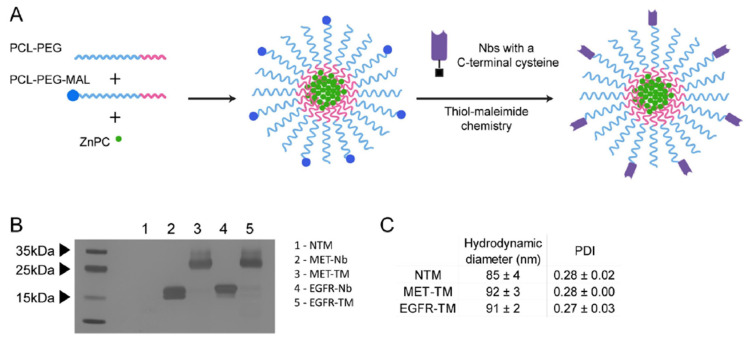
Nbs targeting MET or EGFR were successfully conjugated to zinc phthalocyanine-loaded micelles. (**A**) Schematic of Nb-targeted micelles prepared via thiol-maleimide chemistry. (**B**) SDS-PAGE with silver staining of (1) Non-targeted micelles, (2) MET-Nb, (3) MET-targeted micelles, (4) EGFR-Nb, (5) EGFR-targeted micelles before the centrifugation step. (**C**) Size and polydispersity of non-targeted micelles, MET-targeted micelles and EGFR-targeted micelles (N = 3). Adapted with permission from [211]. Copyright 2024, Elsevier.

**Figure 12 nanomaterials-15-00158-f012:**
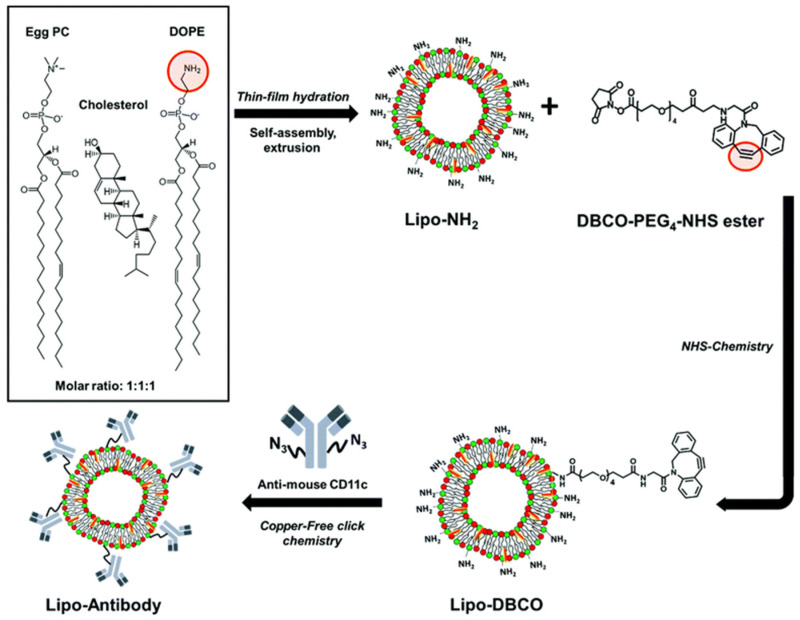
Schematic illustration of step-by-step liposome synthesis, surface functionalization and site-selective Ab attachment. Adapted with permission from [294]. Copyright 2020, Royal Society of Chemistry.

**Figure 13 nanomaterials-15-00158-f013:**
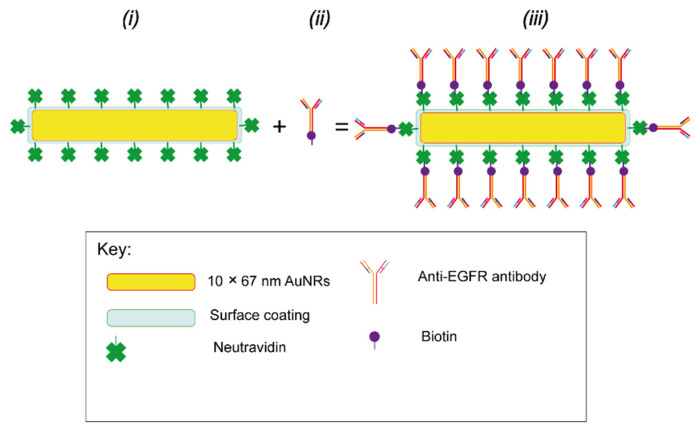
Schematic representation of the functionalization of neutravidin-coated gold nanorods with biotinylated anti-EGFR Abs. (i) Neutravidin-coated 10 nm × 67 nm gold nanorods, (ii) biotinylated 5.2 nm × 5.2 nm anti-EGFR Abs, (iii) Biotinylated anti-EGFR Abs-coated neutravidin gold nanorods. Adapted with permission from [308]. Copyright 2021, MDPI.

**Figure 14 nanomaterials-15-00158-f014:**
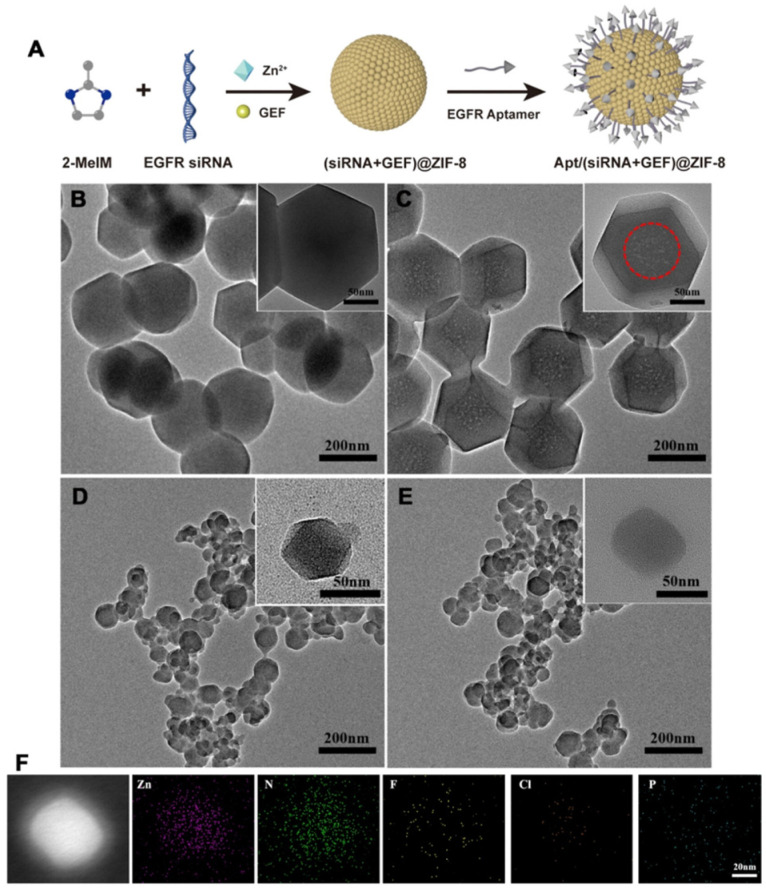
Characterization of multifunctional NPs. (**A**) Schematic illustration of the synthesis procedures for Apt/(siRNA + GEF)@ZIF-8 NPs. (**B**–**E**) TEM analysis of ZIF-8, GEF@ZIF-8, (siRNA + GEF)@ZIF-8 and Apt/(siRNA + GEF)@ZIF-8 NPs, respectively. Red circle highlighting the white dots in (**C**) gives evidence of the successful encapsulation of GEF. (**F**) Dark-field TEM image and elemental mappings of Apt/(siRNA + GEF)@ZIF-8 NPs. Adapted with permission from [321]. Copyright 2022, American Chemical Society.

**Figure 15 nanomaterials-15-00158-f015:**
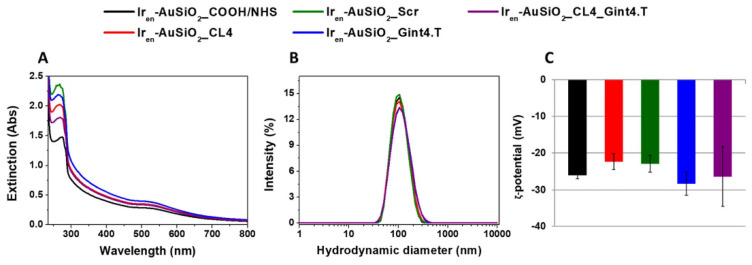
Characterization of aptamer-functionalized NPs at different steps of functionalization. (**A**) UV-Vis spectra. (**B**) Hydrodynamic diameters. (**C**) Zeta potential values of the systems dispersed in water. Adapted with permission from [258]. Copyright 2024, Springer Nature.

**Figure 16 nanomaterials-15-00158-f016:**
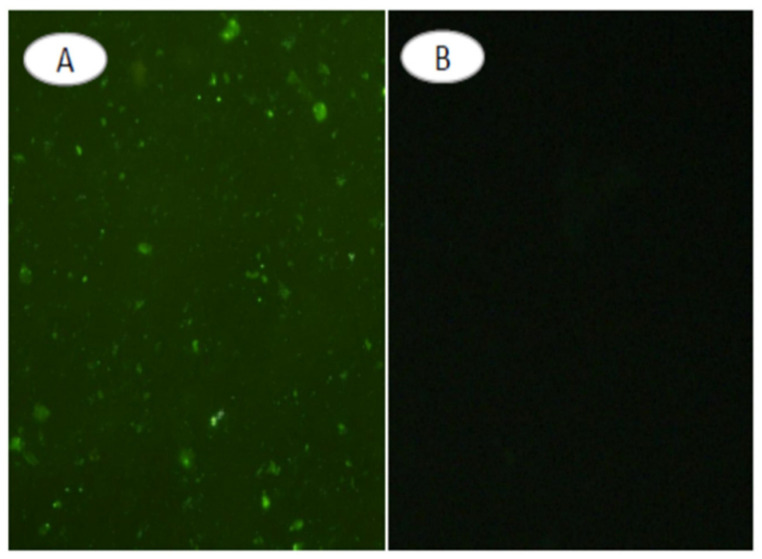
Microscopy images of the secondary Ab labeled with fluorescein bound to mAb-functionalized PLGA NPs. (**A**) mAb-functionalized PLGA NPs were incubated with fluorescein-conjugated secondary Ab. (**B**) mAb-functionalized PLGA NPs without fluorophore. Adapted with permission from [333]. Copyright 2020, Elsevier.

**Figure 17 nanomaterials-15-00158-f017:**
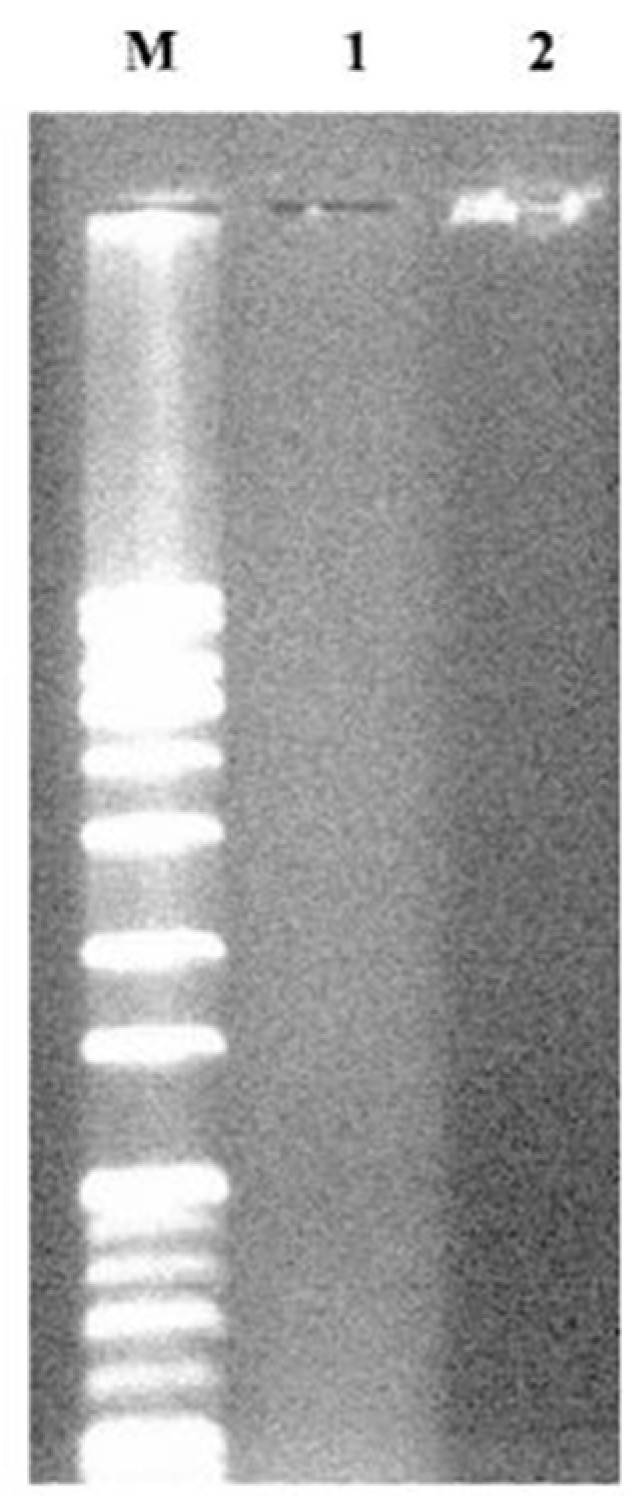
Characterization of aptamer-decorated erlotinib-loaded chitosan NPs. Agarose gel electrophoresis of (1) erlotinib-loaded chitosan NPs and (2) aptamer-decorated erlotinib-loaded chitosan NPs. Adapted with permission from [337]. Copyright 2020, Elsevier.

**Figure 18 nanomaterials-15-00158-f018:**
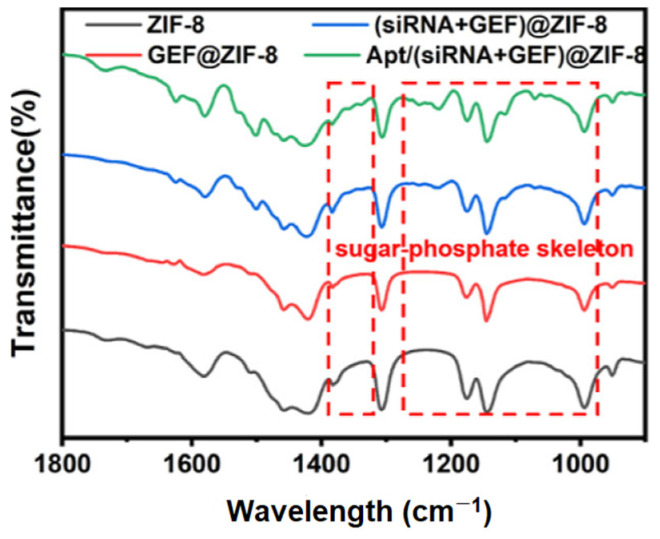
FTIR spectra of the aptamer-functionalized NPs at different steps of functionalization. Adapted with permission from [321]. Copyright 2023, American Chemical Society.

**Figure 19 nanomaterials-15-00158-f019:**
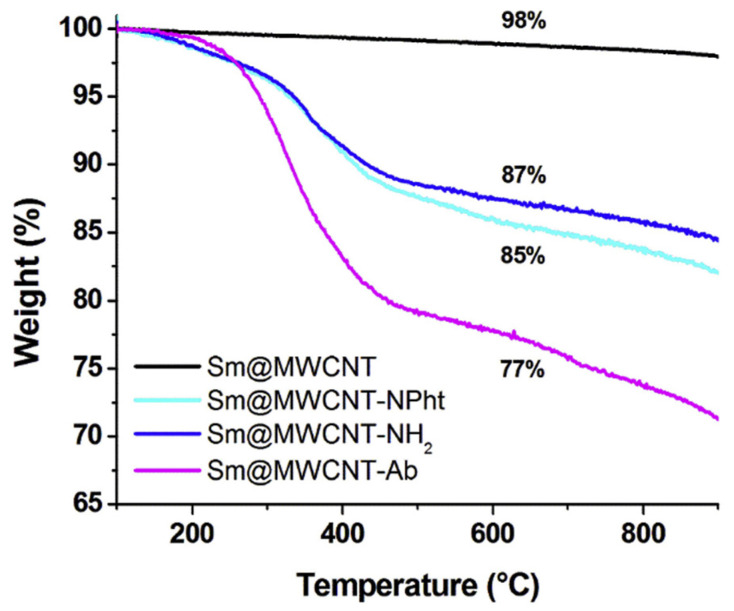
TGA curves of Ab-functionalized NPs at different steps of functionalization. The analyses were performed under N_2_ using a ramp of 10 °C/min. The level of functionalization was calculated from the weight loss values at 650 °C. Adapted with permission from [357]. Copyright 2020, Elsevier.

**Table 1 nanomaterials-15-00158-t001:** Examples of nanocarriers decorated with anti-EGFR Ab fragments.

Ab Fragment	Type of NPs	Applications and Outcomes	References
scFv (husA)	Inorganic (FITC-labelled DOX-loaded MSNPs)	Reduction in viability of EGFR-overexpressing A431, HeLa and MCF-7 cells; higher uptake in EGFR-overexpressing A431 cells than in EGFR-low expressing HEK293 cells; inhibited tumor growth in A431 tumor-bearing mice.	[191]
scFv	Inorganic (Fe_3_O_4_/AuNPs)	MRI bioprobe for detection and treatment of NSCLC; higher uptake in EGFR-overexpressing SPC-A1 cells than in EGFR-low expressing H69 cells; in vivo tumor accumulation in SPC-A1 tumors and not in H69 tumors.	[192]
scFv	Organic (si-RNA-loaded engineered exosomes)	Inhibition of lung cancer brain metastasis; higher uptake in EGFR-overexpressing lung cancer cells PC9; efficient siRNA delivery across blood–brain barrier (BBB) in tumor-bearing mice.	[193]
scFv	Inorganic-organic (superparamagnetic iron oxide NPs (SPIONs) coated with PEG and chitosan)	siRNA delivery into TNBC cells; higher cellular uptake (1.5x) in EGFR-overexpressing MDA-MB-231 cells; protection of siRNA and 69.4% in vitro transfection efficiency.	[194]

**Table 2 nanomaterials-15-00158-t002:** Anti-EGFR targeting ligands used for conjugation to nanocarriers.

Targeting Ligand	Advantages	Limitations	Structure and MW	Kd	References
EGF	Small size enabling high tumor penetration; smaller final size of the system obtained; high natural affinity for EGFR; low cytotoxicity; moderate stability in vitro; stability at physiological conditions and neutral pH; ease of conjugation to nanocarriers; availability via recombinant expression	Expensive production; possible antigenicity issues and immune responses in vivo; prone to proteolysis in vivo	Small protein of 53 amino acids, 6 kDa	2 nM	[99,100,101,102,103,104,105,106,107,108,109,110,111,112]
GE11	Very small size facilitating tumor penetration and diffusion; smaller final size of the system obtained; no mitogenic activity; high chemical stability; ease of production and synthesis; cost-effective manufacture	Lower affinity for EGFR but sufficient for targeting purposes	Small peptide of 12 amino acids, 1.54 kDa	22 nM	[116,117,118,119,120,121,122,123,124,125,126,127,128,129,130,131,132,133,134,135,136,137,138,139,140,141,142,143,144]
Antibodies	Large size allowing increased circulation time and long half-life in vivo; larger final size of the system obtained; possible aggregation of the system observed; possible multivalent receptor binding; highest binding affinity; longest history of usage; possible therapeutic effects through EGFR pathway inhibition	Poor tissue penetration due to bulky nature and large size; high immunogenicity, poor stability, challenging control over oriented conjugation	Y-shaped protein consisting of two heavy and two light chains, 150 kDa	0.05–20 nM	[152,153,154,155,156,157,158,159,160,161,162,163,164,168,170,171,172,173,174,175,176,177,178]
Ab fragments	Smaller size than full Abs facilitating higher tumor penetration; smaller final size of the system obtained; reduced immunogenicity due to lack of Fc regions; easier and cheaper production compared to full mAbs; retained high affinity of the binding regions; high loading capacity; controlled orientation	Generally lower binding affinity than full Abs; faster renal clearance and reduced circulation times (that can be overcome with conjugation to NPs)	F(ab’)2: two antigen-binding sites joined at the hinge region through disulfide bonds, 110 kDa Fab’: reduced F(ab’)2 fragment, containing a free sulfhydryl group, 55 kDa Fab: monovalent fragments composed of the V_H_, C_H1_, V_L_ and C_L_ regions, 50 kDa	1–10 nM	[191,192,193,194,195,196,197,198,199,200,201]
Nanobodies	Small size enabling high tumor penetration; smaller final size of the system obtained; high stability being resistant to denaturation and proteolysis; low immunogenicity; good solubility; binding to unique epitopes; versatile functionalization; easy and cost-effective production	Short half-life and rapid clearance from circulation (that can be overcome with conjugation to NPs); limited applications and clinical studies compared to full mAbs and Ab fragments	Heavy-chain variable domains (V_HH_) of camelid and shark Abs, 12–15 kDa	2–25 nM	[209,210,211,212,213,214]
Affibodies	Small size enabling high tumor penetration; smaller final size of the system obtained; high thermal and chemical stability; low immunogenicity due to Ab mimics and engineered protein scaffold nature; good solubility; large number of libraries available; versatile functionalization; ease of production and synthesis	Short half-life, rapid clearance from circulation (that can be overcome with conjugation to NPs); limited applications and clinical studies compared to full mAbs and Ab fragments	Three-helix bundle formed by 58 amino acids, 6 kDa	2.8 nM	[224,225,226,227,228,229,230,231,232]
Aptamers	Small size and flexibility enabling excellent tumor penetration; smaller final size of the system obtained; lower electrostatic adsorption to nanocarriers compared to protein ligands; high binding affinity, high thermal and chemical stability; versatile binding capability; low immunogenicity and toxicity in vivo; large number of libraries available; versatile functionalization and chemical modification; easy and cost-effective production; minimal batch-to-batch variations	Short half-life, rapid clearance from circulation (that can be overcome with conjugation to NPs); susceptible to nuclease degradation without chemical modifications; limited applications and clinical studies compared to full mAbs and Ab fragments	Short single-stranded DNA or RNA molecules with specific 3D architectures, 6–25 kDa	2.4 nM	[255,256,257,258,259,260,261,262,263,264,265,266]

**Table 3 nanomaterials-15-00158-t003:** Examples of nanocarriers functionalized with anti-EGFR ligands using carbodiimide chemistry.

Targeting Ligand	NP Type	Main Observations	References
C225	Multilayered polyelectrolyte capsules, optically encoded with fluorescent quantum dots	~87% Ab coupling efficiency; random Ab orientation reduced steric hindrance while maintaining selective interaction with targets.	[269]
GE11 peptide	SPIONs	Complete peptide conjugation confirmed via mass spectroscopy analysis of the filtrate; maximum cellular uptake was achieved in 24 h; internalization was proportional to EGFR expression of the cell line tested.	[137]
EGF	Carboplatin-loaded alginate poly(amidoamine) (PAMAM) hybrid NPs	Success of EGF conjugation confirmed in 2D and 3D in vitro setting; final NP size after conjugation was smaller than 400 nm; conjugated EGF at the surface increased with respect to its concentration in the reaction media; EGF-conjugated platforms exerted the highest therapeutic potential in vivo.	[111]
Anti-EGFR aptamers	Ganoderenic acid D-loaded PEGylated graphene oxide-based carrier	Simultaneous conjugation of aptamers and FITC via amino groups; successful anti-tumor effects of targeted platform were confirmed in vitro and in vivo.	[270]
Anti-EGFR Afbs	Gadolinium-encapsulated carbonaceous dots	Higher uptake in EGFR-overexpressing cells and tumor xenografts was confirmed by fluorescence and T1-weighted MRI results; efficient clearance of targeted platform by the renal system was achieved.	[224]
Anti-EGFR ScFv	Human serum albumin-coupled, FITC-labeled and DOX-loaded mesoporous silica NPs	In vitro and in vivo anti-tumor activity and safety of the targeted platform was verified; prolonged circulation, targeted accumulation, and enzyme/pH-responsive drug release properties were confirmed.	[191]

**Table 4 nanomaterials-15-00158-t004:** Conjugation of anti-EGFR targeting ligands to AuNPs through gold-thiol chemisorption.

Targeting Ligand	Application	Main Observations	References
EGF	Development of a new contrast agent for early stage tumors based on AuNPs and gadopentetic acid	Au@Gd-EGF exhibited significant MRI signal intensity for diagnostic applications, allowing for high specificity and sensitivity; the targeted nanocontrast agent showed good biocompatibility and low cytotoxicity in vitro.	[286]
C225	Systematic comprehensive characterization and stability assessment of a targeted nanocomplex with high potential for biomedical applications	After 24 months manufacturing, decoupling Ab from AuNPs was not observed, suggesting irreversible immobilization; efficient EGFR binding and induced tumor cell death due to apoptosis were confirmed.	[287]
Anti-EGFR aptamer (U2)	Development of a novel brain-targeting complex for glioblastoma multiforme therapy	The U2-AuNPs inhibited the proliferation and invasion of EGFR-overexpressing cells; the targeted platform allowed to cross the blood–brain barrier and prolonged survival time of glioblastoma-bearing mice.	[266]
Anti-EGFR aptamer + Anti-EGFR Ab	Development of multi-functionalized probe for detection of EGFR-positive cancer cells	Dual targeted platform showed higher target specificity to EGFR-positive cancer cells, compared to Apt- or Ab-functionalized probes; main benefits of the platform are its potential to facilitate the detection of binding to cell surface markers with low expression levels.	[265]

**Table 5 nanomaterials-15-00158-t005:** Bioconjugation strategies for grafting anti-EGFR ligands to nanocarriers.

Bioconjugation Chemistry	Mechanism	Advantages	Limitations	References
Carbodiimide chemistry	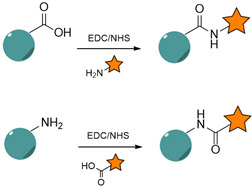	Easy to perform and accessible, generally no need for chemical modification, high stability, high conjugation efficiency	Sensitivity to pH, inability to control orientation, possible competition by extraneous carboxyl or amine groups	[111,137,191,224,269,270,271]
Schiff base reaction	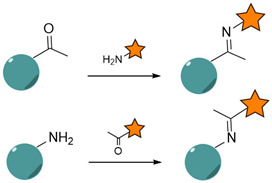	Mild reaction conditions, possibility of oriented conjugation	Specific functional groups and modifications required, sensitivity to pH	[173,275,276,277]
Thiol-maleimide chemistry	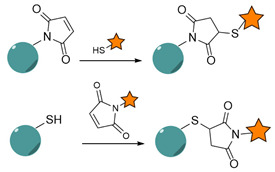	Fast and efficient reactions, mild reaction conditions, higher selectivity compared to carbodiimide chemistry, possibility of reversible linkages	Inability to control orientation, introduction of linkers or modifications required, possibility to hinder biological activity of ligands after reduction, sensitivity of thioether linkages to endogenous reducing potentials	[116,158,211,231,283]
Dative chemistry	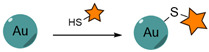	Direct conjugation, no modification or introduction of functional moieties required	Sensitivity to pH, oxidation or replacement by similar molecules, weaker bonds than covalent linkages	[265,266,286,287]
Click chemistry	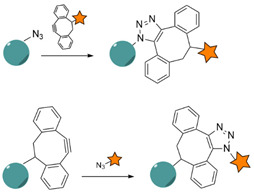	High selectivity, mild reaction conditions, high yields, favorable reaction rates, irreversible chemical linkages, no complex purification required, biorthogonal reactions, possibility of oriented conjugation	Toxicity of Cu(I) catalyst in CuAAC	[157,227,294,296,298,299,300,301]
Interaction by adapter molecules	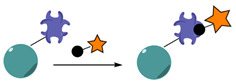	Strongest non-covalent interaction, oriented conjugation, modifications required, high stability	Non-specific binding mediated by avidin, possible aggregation and big sizes due to the presence of the bulky protein	[308,309,310]
Electrostatic interaction	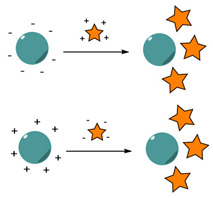	No modification required, cost-effective synthesis and post-functionalization	Less robust and more prone to degradation than covalent linkages, sensitive to experimental conditions, non-oriented conjugation	[101,319,320,321,322]
Fc-binding receptors mediated conjugation	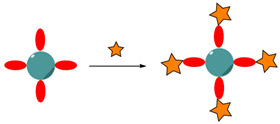	Oriented conjugation, no modifications required, mild reaction conditions	Weaker nature compared to covalent bonds, potential immune activation due to the presence of Fc regions	[323,325]

**Table 6 nanomaterials-15-00158-t006:** Examples of nanocarriers functionalized with anti-EGFR ligands and characterized via electrophoretic techniques.

Targeting Ligand	NP Type	Electrophoretic Method	References
Anti-EGFR Abs	AuNPs	SDS-PAGE	[168]
Anti-EGFR Abs	PLGA NPs	SDS-PAGE	[338]
Anti-EGFR aptamers	Chitosan NPs	Agarose gel electrophoresis	[337]
Anti-EGFR aptamers	Liposomes	Agarose gel electrophoresis	[339]

**Table 7 nanomaterials-15-00158-t007:** Examples of nanocarriers functionalized with anti-EGFR ligands and characterized via protein-based techniques.

Targeting Ligand	NP Type	Result of Quantification	References
C225	Chitosan/hyaluronic acid NPs	Bradford assay revealed a degree of conjugation of 81.5%.	[344]
C225	Sialic acid-coated chitosan NPs	Bradford assay revealed a degree of conjugation of 72.9% ± 3.5% for chitosan NPs, and of 63.2% ± 2.1% for sialic acid-coated chitosan NPs. The second is lower probably due to the presence of sialic acid residues along with C225.	[345]
C225	Silica-coated AuNPs	BCA assay performed on the supernatant revealed 91% ± 6% of Abs conjugated to the NP surface.	[346]
GE11	DOX-loaded extracellular vesicles	BCA assay was performed to measure the total protein content of EVs and used to add equivalent EVs to EGFR-overexpressing cell lines (uptake value to be compared with the uptake by EGFR-negative cell lines).	[347]

**Table 8 nanomaterials-15-00158-t008:** Examples of nanocarriers functionalized with anti-EGFR ligands and characterized via spectroscopic techniques.

Targeting Ligand	NP Type	Characterization Technique	Analyzed Components	References
Anti-EGFR Ab (Nm)	AuNPs	FTIR	Appearance of a small peak at 3300 cm^−1^, related to the region of amide A formed by the crosslinking between the Abs and the NHS linker causing a N-H stretch, was detected. The band at 1600–1700 cm^−1^ revealed the presence of amide-I band related to the amide C=O stretching vibrations. The band at 2500–2600 cm^−1^ indicated the presence of S-H in the conjugates coming from the thiol-gold bond.	[168]
Anti-EGFR aptamer	ZIF-8 NPs	FTIR	Special adsorption of the sugar-phosphate skeleton of siRNA and aptamers related with the peaks from 1300 to 900 and 1320 to 1380 cm^−1^ was observed.	[321]
GE11	DOX-loaded polymeric conjugates	^1^H NMR	Additional peaks corresponding to the GE11 peptide appeared at 6.98–6.63 ppm, revealing successful conjugation of the peptide to the NP surface.	[116]
C225	Chitosan NPs	XPS	Percentages of N 1s, O 1s and C 1s in the targeted system were 9.74%, 22.54% and 67.74%, respectively, whereas for the non-targeted counterpart, these were of 1.04%, 15.19% and 83.77%. The higher percentage nitrogen in the targeted platform indicated the presence of a large number of nitrogen atoms (N = 1732) in the Ab molecules.	[354]
C225	Chitosan/hyaluronic acid NPs	XPS	Atomic percentage of N exhibited a significant increase from 3.53% in the non-targeted system to 7.74% in the targeted system, together with the apparition of a sulfur peak (0.13%). The phosphorous peak due to the crosslinker used during the ionic gelation of NPs was significantly decreased in the case of the targeted system, further indication of NP functionalization.	[344]

**Table 9 nanomaterials-15-00158-t009:** Characterization methods to evaluate the conjugation of anti-EGFR targeting ligands at the NP surface.

Characterization Technique	Type of Information	Purpose	Advantages	Limitations	References
Dynamic light scattering	Qualitative	Measurement of hydrodynamic size, polydispersity and surface charge	Quick and easy to perform, any types of NPs	Often not reliable or informative enough, no quantification possible	[256,258,328,329,330,331]
Fluorescence experiments	Quantitative	Measurements of the density of ligands via indirect or direct fluorescence reading	Generally high sensitivity; precise quantification via comparison with calibration standards	Modification of ligand with fluorophore or use of fluorophore-tagged secondary ligand is required, thorough purification to remove excess of fluorophore-tagged ligand is needed	[173,332,333]
Electrophoretic techniques	Qualitative	Separation and visualization of protein or DNA-based ligands depending on their migration rates	Visual method	Mainly used for protein and oligonucleotide ligands, quantification is generally difficult to perform	[168,337,338,339]
Protein-based assays	Quantitative	Measurements of the density of protein ligands by absorbance readings	Sensitive methods, precise quantification possible, fast procedures	Only protein-based ligands can be investigated, sensitivity to reducing agents or detergents is observed	[344,345,346,347]
Spectroscopy techniques	Qualitative	Determine the nature of surface functional groups, identify the elemental composition and investigate molecular structure of ligands	different types of NPs can be investigated	Quantification not possible (except in some techniques by using calibration standards), high densities and thorough purification are required in some techniques	[116,168,321,344,354]
Thermogravimetric analysis	Qualitative /quantitative	Measure mass change depending on temperature to determine the surface coverage	Quantification is possible, simple procedure	Usable only to investigate surface of inorganic NPs, high amount of material is generally required for analysis	[357,358]

## Data Availability

We do not present original data in the manuscript as it is a review article.

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
