# Peer review of "Surface Functionalization of Nanocarriers with Anti-EGFR Ligands for Cancer Active Targeting"

_nanomaterials, 2025, doi:10.3390/nano15030158_

Round 1

Reviewer 1 Report

Comments and Suggestions for Authors

This in-depth review is very well written and presents a topic that is pertinent to the study of nanotherapeutics. The writing is polished, and I do not have any major concerns. 

Author Response

Comment 1: This in-depth review is very well written and presents a topic that is pertinent to the study of nanotherapeutics. The writing is polished, and I do not have any major concerns. 

Response 1: We thank the reviewer for his/her appreciation of our manuscript.

Reviewer 2 Report

Comments and Suggestions for Authors

Active targeting offers a more specific and selective method compared to passive targeting. It enhances the accumulation of drugs in tumor tissues while minimizing off-target effects. The development of anti-EGFR nanocarriers represents a promising strategy for improving the efficacy of cancer therapies. By leveraging active targeting mechanisms and optimizing conjugation strategies, these systems can enhance drug accumulation in tumor tissues, potentially leading to better treatment outcomes. An interesting work. I suggest a revision.

1. The presentation depicted in Figure 1-3 could benefit from a more professional appearance. The author is encouraged to consult high-level relevant charts for guidance.

2. The author is advised to consult a range of relevant literature. For example, “Cancer mortality is reduced when cases are detected and treated at an early stage of development. (DOI: 10.1002/INMD.20220005)”; “Nanoparticles (NPs) have attracted interest due to their unique features (DOI: 10.1016/j.jcis.2024.08.100). They are very small colloidal systems (usually from 5 to 200 nm) whose morphology and properties depend on their compositions. (DOI: 10.1049/nbt2.12018)”; “However, there are several limitations with regards to passive targeting, including non-specific NPs distribution, non-universal existence of the EPR effect and different permeability of blood vessels across various tumors. (DOI:10.1007/978-981-97-3545-7_7)”

3. Exploring the use of anti-EGFR nanocarriers in combination with other treatment modalities, such as immunotherapy, may enhance therapeutic efficacy.

4. The choice of target biomarker is critical. The epidermal growth factor receptor (EGFR) is one of the most frequently overexpressed receptors in many cancer types, making it an ideal target for therapeutic interventions. However, it continues to encounter considerable challenges in clinical practice. A comprehensive examination of these issues is warranted.

5. In “Table 2. Anti-EGFR targeting ligands used for conjugation to nanocarriers.” please provide a more detailed elaboration regarding both the advantages and limitations. For instance, consider discussing specific aspects such as particle size.

Author Response

Comment 1: The presentation depicted in Figure 1-3 could benefit from a more professional appearance. The author is encouraged to consult high-level relevant charts for guidance.

Response 1: The Figure 1-3 were replaced with more professional versions which were made with BioRender.com.

Comment 2: The author is advised to consult a range of relevant literature. For example, “Cancer mortality is reduced when cases are detected and treated at an early stage of development. (DOI: 10.1002/INMD.20220005)”; “Nanoparticles (NPs) have attracted interest due to their unique features (DOI: 10.1016/j.jcis.2024.08.100). They are very small colloidal systems (usually from 5 to 200 nm) whose morphology and properties depend on their compositions. (DOI: 10.1049/nbt2.12018)”; “However, there are several limitations with regard to passive targeting, including non-specific NPs distribution, non-universal existence of the EPR effect and different permeability of blood vessels across various tumors. (DOI: 10.1007/978-981-97-3545-7_7)”.

Response 2:

The reference DOI: 10.1002/INMD.20220005 (ref 30 in the revised review manuscript) was added in the introduction section where the ability of the NPs to improve penetration and accumulation in tumors was discussed (lines 68-72). The innovative study is mentioned as an example of the development of intelligent drug carriers, which respond to signal changes from lesion sites and allow to achieve precise target of loaded therapeutic agents, improving the efficiency and final outcome of the treatment.

The reference DOI: 10.1016/j.jcis.2024.08.100 (ref 22 in the revised review manuscript) was added in the introduction section, where the topic of multi-drug resistance (MDR) is addressed (line 59). The study is mentioned as example of one of the ways in which nanocarriers can decrease MDR. The computer-aided based “triadic” drug self-delivery system acts on significantly reducing the expression of the P-glycoprotein (P-gp), a drug efflux pump often overexpressed in drug-resistant cells.

The reference DOI: 10.1049/nbt2.12018 (ref 5 in the revised review manuscript) was added in the introduction section, where a paragraph describing the types of NPs used for biomedical applications was incorporated (lines 37-52).

The reference DOI: 10.1007/978-981-97-3545-7_7 unfortunately was not accessible. We tried to open it in multiple ways, even creating a Spring Nature account as suggested, but the text was never available.

Comment 3: Exploring the use of anti-EGFR nanocarriers in combination with other treatment modalities, such as immunotherapy, may enhance therapeutic efficacy.

Response 3: Literature reports on the combination of EGFR targeting and immunotherapy were analyzed, and a corresponding paragraph was added in the Introduction section (lines 111-132).

Comment 4: The choice of target biomarker is critical. The epidermal growth factor receptor (EGFR) is one of the most frequently overexpressed receptors in many cancer types, making it an ideal target for therapeutic interventions. However, it continues to encounter considerable challenges in clinical practice. A comprehensive examination of these issues is warranted.

Response 4: The primary challenges associated with EGFR targeting and the gaps in its clinical translation were examined and discussed (lines 221-243). Additionally, potential solutions and future directions were also proposed.

Comment 5: In “Table 2. Anti-EGFR targeting ligands used for conjugation to nanocarriers.” please provide a more detailed elaboration regarding both the advantages and limitations. For instance, consider discussing specific aspects of particle size.

Response 5: Table 2 was modified and expanded introducing additional details regarding the advantages and limitations of each class of targeting agents, focusing in particular on the size of the obtained targeted systems. In the column “structure and MW”, the molecular weight of each class of ligand is specified.

Reviewer 3 Report

Comments and Suggestions for Authors

The authors reports an extensive review on surface functionalization of nanoparticles with anti-EGFR ligands for tumor targeting. The review majorly focuses on reports in last 5 years and explores different Anti-EGFR ligands, conjugation strategies and characterization methods. The authors did a great job compiling some of the key advantages, differences or shortcomings in the form of tables. Further, a balanced amount of introduction and background is provided for each section to help readers from different chemical or biological backgrounds. 

Below are some suggestions to improve this article: 

1. Authors should expand the figure legends to properly explain the figures. 

2. Line 47-48: Citation should be provided to identify size for RES escape.

3. Line 334:  full form of MrNV should be provided. 

4. Line 353: Abs should be called 'large' proteins to differentiate from the nomenclature of light and heavy chains. Also Ab size is "approximately" 150 nm. 

5. Line 489-493: Can authors add comments on the possible mechanism behind preferred coupling of fragment mAbs with lipid NPs and full mAbs with polymeric/organic/inorganic NPs?

6. Table 1: HEK293 cells instead of just 293 cells

7. Line 601: Typo as it should be "in vivo"

8. The authors mentioned that RNA aptamers has very low half lives in plasma, yet there are several reports of NPs with RNA aptamers. Can authors add comments on possible reasons for exploring RNA aptamers despite low stability?

9. Line 663: Typo - RNA aptamer-functionalized NPs

10. Line 735: Typo - Linker

11. Line 815-816: Full form of IEDDA is incorrect and abbreviation is not written at the right spot.

12. line 969: should be labelled secondary Ab

13. Authors should add the following reports as well:    

https://doi.org/10.1038/s41598-024-80879-0

https://doi.org/10.1016/j.jconrel.2024.05.036

14. Authors should consider adding 1-2 figures to highlight a few of most impactful reports for key sections. 

Author Response

Comment 1:  Authors should expand the figure legends to properly explain the figures.

Response 1: A more detailed legend was added for all the figures present in the manuscript.

Comment 2: Line 47-48: Citation should be provided to identify size for RES escape.

Response 2: The respective references for the impact of NPs size on RES escape were added [Zhang, M.; Gao, S.; Yang, D.; Fang, Y.; Lin, X.; Jin, X.; Liu, Y.; Liu, X.; Su, K.; Shi, K. Influencing Factors and Strategies of Enhancing Nanoparticles into Tumors in Vivo. Acta Pharm. Sin. B 2021, 11, 2265–2285, doi:10.1016/j.apsb.2021.03.033.

Hoshyar, N.; Gray, S.; Han, H.; Bao, G. The Effect of Nanoparticle Size on in Vivo Pharmacokinetics and Cellular Interaction. Nanomed. 2016, 11, 673–692, doi:10.2217/nnm.16.5. Grandhi, S.; Al-Tabakha, M.; Avula, P.R. Enhancement of Liver Targetability through Statistical Optimization and Surface Modification of Biodegradable Nanocapsules Loaded with Lamivudine. Adv. Pharmacol. Pharm. Sci. 2023, 2023, 8902963, doi:10.1155/2023/8902963. Blanco, E.; Shen, H.; Ferrari, M. Principles of Nanoparticle Design for Overcoming Biological Barriers to Drug Delivery. Nat. Biotechnol. 2015, 33, 941–951, doi:10.1038/nbt.3330.] (in the revised review: ref 25-28].

Comment 3: Line 334: full form of MrNV should be provided.

Response 3: The abbreviation MrNV was explained and full form was provided at line 419-420.

Comment 4: Line 353: Abs should be called “large” proteins to differentiate from the nomenclature of light and heavy chains. Also Ab size is “approximately” 150 nm.

Response 4: The term “large” was inserted in the text replacing “heavy”, that might be redundant next to the description of “heavy” chains of Abs. However, the information regarding weight and size of Abs was kept, since the molecular weight of antibodies is approximately 150 kDa, which corresponds to typical dimensions of around 14 nm x 8 nm x 3 nm. An additional reference for the molecular weight of Abs was added [Sandin, S.; Öfverstedt, L.-G.; Wikström, A.-C.; Wrange, Ö.; Skoglund, U. Structure and Flexibility of Individual Immunoglobulin G Molecules in Solution. Structure 2004, 12, 409–415, doi:10.1016/j.str.2004.02.011] (in the revised review: ref 147).

Comment 5: Line 489-493: Can authors add comments on the possible mechanism behind preferred coupling of fragment mAbs with lipid NPs and full mAbs with polymeric/organic/inorganic NPs?

Response 5: The authors of the meta-analysis did not discuss potential mechanisms, stating that the results should be taken with precautions due to the relatively limited size sample analyzed. The primary outcome of the meta-analysis was the significant enhancement in tumor accumulation achieved by targeted NPs, including full mAb-NPs and fragment-NPs, compared to non-targeted NPs, regardless of the NP type used. Furthermore, full mAbs and fragments exhibited comparable NPs tumor uptake. Nevertheless, for both polymeric and organic/inorganic NPs, grafting of full mAbs resulted in an improved NP uptake compared to fragment-NPs, regardless of the overall size of the conjugates. Conversely, lipidic NPs showed greater uptake with fragment-NPs, which correlated positively with the system’s size. To our understanding, this phenomenon could potentially be attributed to specific physicochemical properties of the NPs, such as size, flexibility, and clearance rates of the final targeted-delivery system. As reported in the review, fragment mAbs exhibit smaller sizes, which facilitate enhanced tumor penetration [Gil, D.; Schrum, A.G. Strategies to Stabilize Compact Folding and Minimize Aggregation of Antibody-Based Fragments. Adv. Biosci. Biotechnol. Print 2013, 4, 73–84, doi:10.4236/abb.2013.44A011]. Moreover, as opposed to full mAbs, their conjugation to nanocarriers is characterized by a more controlled orientation [Greene, M.K.; Richards, D.A.; Nogueira, J.C.F.; Campbell, K.; Smyth, P.; Fernández, M.; Scott, C.J.; Chudasama, V. Forming Next-Generation Antibody–Nanoparticle Conjugates through the Oriented Installation of Non-Engineered Antibody Fragments. Chem. Sci. 2017, 9, 79–87, doi:10.1039/C7SC02747H]. Hypothetically, lipid NPs, due to their relatively small and flexible nature, could work synergistically with smaller fragment mAbs to enhance tumor uptake and accumulation. According to the meta-analysis, fragment mAbs-lipid NPs with smaller sizes demonstrated higher targeting efficiency. This combination leverages both active targeting, enabled by the specific targeting agent, and passive targeting, facilitated by improved extravasation into tumor tissues via EPR effect. On the other hand, full mAbs, being larger and bulkier, contribute to an increased overall size of the delivery system. Polymeric and inorganic NPs, typically larger and less flexible than lipid NPs, are less capable of exploiting the EPR effect for passive targeting. However, these NPs are known for their high stability [Lu, H.; Zhang, S.; Wang, J.; Chen, Q. A Review on Polymer and Lipid-Based Nanocarriers and Its Application to Nano-Pharmaceutical and Food-Based Systems. Front. Nutr. 2021, 8, 783831, doi:10.3389/fnut.2021.783831] and may achieve greater tumor uptake when functionalized with high affinity ligands, such as full mAbs.

Comment 6: Table 1: HEK293 cells instead of just 293 cells.

Response 6: The appropriate correction was made in Table 1.

Comment 7: Line 601: Typo as it should be “in vivo”.

Response 7: The correction was made in text by replacing “in vitro” with “in vivo” (line 714).

Comment 8: The authors mentioned that RNA aptamers have very low half lives in plasma, yet there are several reports of NPs with RNA aptamers. Can authors comment on possible reasons for exploring RNA aptamers despite low stability?

Response 8: Despite the low stability and short half-life in plasma, RNA aptamers possess several advantages over other targeting ligands and the potential to overcome limitations linked to their stability. First, aptamers have been shown to assume more diverse and intricate three-dimensional structures than the DNA counterparts, enabling a higher number of possible conformations [Zhu, Q.; Liu, G.; Kai, M. DNA Aptamers in the Diagnosis and Treatment of Human Diseases. Mol. Basel Switz. 2015, 20, 20979–20997, doi:10.3390/molecules201219739. Orava, E.W.; Cicmil, N.; Gariépy, J. Delivering Cargoes into Cancer Cells Using DNA Aptamers Targeting Internalized Surface Portals. Biochim. Biophys. Acta 2010, 1798, 2190–2200, doi:10.1016/j.bbamem.2010.02.004]. Several methods to enhance the stability and reduce nuclease degradation have been thoroughly investigated, among which chemical modifications, including substitution of the natural hydroxyl group at the 2’ position of the RNA bases with a fluorine, protection of the 5’ and 3’ ends of RNA aptamers through capping, or circularization of RNA [Kratschmer, C.; Levy, M. Effect of Chemical Modifications on Aptamer Stability in Serum. Nucleic Acid Ther. 2017, 27, 335–344, doi:10.1089/nat.2017.0680. Fallah, A.; Imani Fooladi, A.A.; Havaei, S.A.; Mahboobi, M.; Sedighian, H. Recent Advances in Aptamer Discovery, Modification and Improving Performance. Biochem. Biophys. Rep. 2024, 40, 101852, doi:10.1016/j.bbrep.2024.101852. Chen, Z.; Luo, H.; Gubu, A.; Yu, S.; Zhang, H.; Dai, H.; Zhang, Y.; Zhang, B.; Ma, Y.; Lu, A.; et al. Chemically Modified Aptamers for Improving Binding Affinity to the Target Proteins via Enhanced Non-Covalent Bonding. Front. Cell Dev. Biol. 2023, 11, doi:10.3389/fcell.2023.1091809] (in the revised review: ref 244-246). Most importantly, conjugation of RNA aptamers, as of other ligands and drugs, to the surface of nanocarriers protects the ligands from enzymatic degradation by shielding them from the extracellular environment and prolonging the half-life in vivo [Fu, Z.; Xiang, J. Aptamer-Functionalized Nanoparticles in Targeted Delivery and Cancer Therapy. Int. J. Mol. Sci. 2020, 21, 9123, doi:10.3390/ijms21239123. Sun, H.; Zhu, X.; Lu, P.Y.; Rosato, R.R.; Tan, W.; Zu, Y. Oligonucleotide Aptamers: New Tools for Targeted Cancer Therapy. Mol. Ther. - Nucleic Acids 2014, 3, e182, doi:10.1038/mtna.2014.32. Rabiee, N.; Chen, S.; Ahmadi, S.; Veedu, R.N. Aptamer-Engineered (Nano)Materials for Theranostic Applications. Theranostics 2023, 13, 5183–5206, doi:10.7150/thno.85419. Shanaa, O.A.; Rumyantsev, A.; Sambuk, E.; Padkina, M. In Vivo Production of RNA Aptamers and Nanoparticles: Problems and Prospects. Molecules 2021, 26, 1422, doi:10.3390/molecules26051422] (in the revised review: ref 240-243).

Comment 9: Line 663: Typo – RNA aptamer-functionalized NPs

Response 9: The correction was made in the text by replacing RNA-functionalized NPs with RNA aptamers-functionalized NPs (line 791).

Comment 10: Line 735: Typo – Linker.

Response 10: The correction of the typo was made in the text (line 871).

Comment 11: Line 815-816: Full form of IEDDA is incorrect and abbreviation is not written at the right spot.

Response 11: Full form of IEDDA was corrected to the right version “inverse electron-demand Diels-Alder reaction”, followed immediately by the abbreviation (IEDDA) (line 959).

Comment 12: Line 969: should be labelled secondary Ab.

Response 12: The appropriate correction was made in the text, from “secondary labelled Ab” to “labelled secondary Ab” (line 1145).

Comment 13:  Authors should add the following reports as well: https://doi.org/10.1038/s41598-024-80879-0; https://doi.org/10.1016/j.jconrel.2024.05.036.

Response 13: The first paper (https://doi.org/10.1038/s41598-024-80879-0) was added in the text at lines 1052-1059, as example of non-covalent electrostatic interaction strategy to graft anti-EGFR mAbs to organic NPs for metastatic colorectal cancer therapy. The second article (https://doi.org/10.1016/j.jconrel.2024.05.036) was not added in the text because, despite its quality and innovative nature, it focuses on targeted platforms for placental dysfunction disorders, such as preeclampsia, characterized by EGFR overexpression. Since our review highlights recent EGFR-targeting systems for cancer active targeting, we opted not to incorporate it into our manuscript.

Comment 14: Authors should consider adding 1-2 figures to highlight a few of most impactful reports for key sections.

Response 14: Several figures from some of the reports mentioned in the paper have been incorporated in various key sections of the review. These figures are appropriately cited in the text and have been adapted from the original sources with permission. 

Round 2

Reviewer 2 Report

Comments and Suggestions for Authors

All raised issues have been addressed and I don't have any additional comment.